# Evaluation of linear regression techniques for atmospheric applications: The importance of appropriate weighting

**Cheng Wu[1,2] and Jian Zhen Yu[3,4,5]**

[1]Institute of Mass Spectrometer and Atmospheric Environment, Jinan University, Guangzhou 510632, China

[2]Guangdong Provincial Engineering Research Center for on-line source apportionment system of air pollution, Guangzhou 510632, China

[3]Division of Environment, Hong Kong University of Science and Technology, Clear Water Bay, Hong Kong, China

[4]Atmospheric Research Centre, Fok Ying Tung Graduate School, Hong Kong University of Science and Technology, Nansha, China

[5]Department of Chemistry, Hong Kong University of Science and Technology, Clear Water Bay, Hong Kong, China

*Corresponding to*: Cheng Wu (wucheng.vip@foxmail.com) and Jian Zhen Yu (jian.yu@ust.hk)

**Abstract**
Linear regression techniques are widely used in atmospheric science, but are often
improperly applied due to lack of consideration or inappropriate handling of
measurement uncertainty. In this work, numerical experiments are performed to
evaluate the performance of five linear regression techniques, significantly extending
previous works by Chu and Saylor. The five techniques are Ordinary Least Square
(OLS), Deming Regression (DR), Orthogonal Distance Regression (ODR), Weighted
ODR (WODR), and York regression (YR). We first introduce a new data generation
scheme that employs the Mersenne Twister (MT) pseudorandom number generator.
The numerical simulations are also improved by: (a) refining the parameterization of
non-linear measurement uncertainties, (b) inclusion of a linear measurement
uncertainty, (c) inclusion of WODR for comparison. Results show that DR, WODR and
YR produce an accurate slope, but the intercept by WODR and YR is overestimated
and the degree of bias is more pronounced with a low $R^2$ XY dataset. The importance
of a properly weighting parameter $\lambda$ in DR is investigated by sensitivity tests, and it is
found that an improper $\lambda$ in DR can lead to a bias in both the slope and intercept
estimation. Because the $\lambda$ calculation depends on the actual form of the measurement
error, it is essential to determine the exact form of measurement error in the XY data
during the measurement stage. If a priori error in one of the variables is unknown, or
the measurement error described cannot be trusted, DR, WODR and YR can provide
the least biases in slope and intercept among all tested regression techniques. For these
reasons, DR, WODR and YR are recommended for atmospheric studies when both X
and Y data have measurement errors.

## 1 Introduction

Linear regression is heavily used in atmospheric science to derive the slope and intercept of XY datasets. Examples of linear regression applications include primary OC (organic carbon) and EC (elemental carbon) ratio estimation (Turpin and Huntzicker, 1995; Lin et al., 2009), MAE (mass absorption efficiency) estimation from light absorption and EC mass (Moosmüller et al., 1998), source apportionment of polycyclic aromatic hydrocarbons using CO and $NO_x$ as combustion tracers (Lim et al., 1999), gas-phase reaction rate determination (Brauers and Finlayson-Pitts, 1997), inter-instrument comparison (Bauer et al., 2009; Cross et al., 2010; von Bobrutzki et al., 2010; Zieger et al., 2011; Wu et al., 2012; Huang et al., 2014; Zhou et al., 2016), inter-species analysis (Yu et al., 2005; Kuang et al., 2015), analytical protocol comparison (Chow et al., 2001; Chow et al., 2004; Cheng et al., 2011; Wu et al., 2016), light extinction budget reconstruction (Malm et al., 1994; Watson, 2002; Li et al., 2017), comparison between modeling and measurement (Petäjä et al., 2009), emission factor study (Janhäll et al., 2010), retrieval of shortwave cloud forcing (Cess et al., 1995), calculation of pollutant growth rate (Richter et al., 2005), estimation of ground level $PM_{2.5}$ from MODIS data (Wang and Christopher, 2003), distinguishing OC origin from biomass burning using $K^+$ as a tracer (Duan et al., 2004) and emission type identification by the EC/CO ratio (Chen et al., 2001).

Ordinary least squares (OLS) regression is the most widely used method due to its simplicity. In OLS, it is assumed that independent variables are error free. This is the case for certain applications, such as determining a calibration curve of an instrument in analytical chemistry. For example, a known amount of analyte (e.g., through weighing) can be used to calibrate the instrument output response (e.g., voltage). However, in many other applications, such as inter-instrument comparison, X and Y (from two instruments) may have comparable degrees of uncertainty. This deviation from the underlying assumption in OLS would produce biased slope and intercept when OLS is applied to the dataset.

To overcome the drawback of OLS, a number of error-in-variable regression models (also known as bivariate fittings (Cantrell, 2008) or total least-squares methods (Markovsky and Van Huffel, 2007) arise. Deming (1943) proposed an approach by

minimizing sum of squares of X and Y residuals. A closed-form solution of Deming
regression (DR) was provided by York (1966). Method comparison work of various
regression techniques by Cornbleet and Gochman (1979) found significant error in OLS
slope estimation when the relative standard deviation (RSD) of measurement error in
"X" exceeded 20%, while DR was found to reach a more accurate slope estimation. In
an early application of the EC tracer method, Turpin and Huntzicker (1995) realized
the limitation of OLS since OC and EC have comparable measurement uncertainty,
thus recommended the use of DR for $(OC/EC)_{pri}$ (primary OC to EC ratio) estimation.
Ayers (2001) conducted a simple numerical experiment and concluded that reduced
major axis regression (RMA) is more suitable for air quality data regression analysis.
Linnet (1999) pointed out that when applying DR for inter-method (or inter-instrument)
comparison, special attention should be paid to the sample size. If the range ratio
(max/min) is relatively small (e.g., less than 2), more samples are needed to obtain
statistically significant results.
In principle, a best-fit regression line should have greater dependence on the more
precise data points rather than the less reliable ones. Chu (2005) performed a
comparison study of OLS and DR specifically focusing on the EC tracer method
application, and found the slope estimated by DR is closer to the correct value than
OLS but may still overestimate the ideal value. Saylor et al. (2006) extended the
comparison work of Chu (2005) by including a regression technique developed by York
et al. (2004). They found that the slope overestimation by DR in the study of Chu (2005)
was due to improper configuration of the weighting parameter, λ. This λ value is the
key to handling the uneven errors between data points for the best-fit line calculation.
This example demonstrates the importance of appropriate weighting in the calculation
of best-bit line for error-in-variable regression model, which is overlooked in many
studies.
In this study, we extend the work by Saylor et al. (2006) to achieve four objectives.
The first is to propose a new data generation scheme by applying the Mersenne Twister
(MT) pseudorandom number generator for evaluation of linear regression techniques.
In the study of Chu (2005), data generation is achieved by a varietal sine function,
which has limitations in sample size, sample distribution, and nonadjustable correlation
($R^2$) between X and Y. In comparison, the MT data generation provides more
flexibility, permitting adjustable sample size, XY correlation and distribution. The
second is to develop a non-linear measurement error parameterization scheme for use
in the regression method. The third is to incorporate linear measurement errors in the
regression methods. In the work by Chu (2005) and Saylor et al. (2006), the relative
measurement uncertainty ($\gamma_{Unc}$) is non-linear with concentration, but a constant $\gamma_{Unc}$
is often applied on atmospheric instruments due to its simplicity. The fourth is to
include weighted orthogonal distance regression (WODR) for comparison.
Abbreviations and symbols used in this study are summarized in Table 1 for quick
reference.
**2    Description of regression techniques compared in this study**
**Ordinary least squares (OLS) method**. OLS only considers the errors in dependent
variables (Y). OLS regression is achieved by minimizing the sum of squares (S) in the
Y residuals (i.e., distance of AB in Fig. S1):

$$S = \sum_{i=1}^{N}(y_i - Y_i)^2 \tag{1}$$

where $Y_i$ are observed Y data points while $y_i$ are regressed Y data points of the
regression line. N represents the number of data points that used for regression.
**Orthogonal distance regression (ODR)**. ODR minimizes the sum of the squared
orthogonal distances from all data points to the regressed line and considers equal error
variances (i.e., distance of AC in Fig. S1):

$$S = \sum_{i=1}^{N}[(x_i - X_i)^2 + (y_i - Y_i)^2] \tag{2}$$

**Weighted orthogonal distance regression (WODR)**. Unlike ODR that considers even
error in X and Y, weightings based on measurement errors in both X and Y are
considered in WODR when minimizing the sum of squared orthogonal distance from
the data points to the regression line (Carroll and Ruppert, 1996) as shown by AD in
Fig. S1:

$$S = \sum_{i=1}^{N}[(x_i - X_i)^2 + (y_i - Y_i)^2/\eta] \tag{3}$$

where $\eta$ is error variance ratio that determines the angle θ shown in Fig. S1.
Implementation of ODR and WODR in Igor Pro (WaveMetrics, Inc. Lake Oswego, OR,
USA) was done by the computer routine ODRPACK95 (Boggs et al., 1989; Zwolak et
al., 2007).
**Deming regression (DR)**. Deming (1943) proposed the following function to minimize
both the X and Y residuals as shown by AD in Fig. S1,
$$S = \sum_{i=1}^{N}[\omega(X_i)(x_i - X_i)^2 + \omega(Y_i)(y_i - Y_i)^2] \tag{4}$$

where $X_i$ and $Y_i$ are observed data points and $x_i$ and $y_i$ are regressed data points.
Individual data points are weighted based on errors in $X_i$ and $Y_i$,
$$\omega(X_i) = \frac{1}{\sigma_{X_i}^2}, \quad \omega(Y_i) = \frac{1}{\sigma_{Y_i}^2} \tag{5}$$

where $\sigma_{X_i}$ and $\sigma_{Y_i}$ are the standard deviation of the error in measurement of $X_i$ and $Y_i$,
respectively. The closed form solutions for slope and intercept of DR are shown in
Appendix A.
**York regression (YR)**. The York method (York et al., 2004) introduces the correlation
coefficient of errors in X and Y into the minimization function.
$$S = \sum_{i=1}^{N}\left[\omega(X_i)(x_i - X_i)^2 - 2r_i\sqrt{\omega(X_i)\omega(Y_i)}(x_i - X_i)(y_i - Y_i) + \omega(Y_i)(y_i - \right.$$

$$\left. Y_i)^2\right]\frac{1}{1-r_i^2} \tag{6}$$

where $r_i$ is the correlation coefficient between measurement errors in $X_i$ and $Y_i$. The
slope and intercept of YR are calculated iteratively through the formulas in Appendix
A.
Summary of the five regression techniques is given in Table S1. It is worth noting that
OLS and DR have closed-form expressions for calculating slope and intercept. In
contrast, ODR, WODR and YR need to be solved iteratively. This need to be taken into
consideration when choosing regression algorithm for handling huge amount of data.
A computer program (Scatter plot; Wu, 2017a) with graphical user interface (GUI) in
Igor Pro (WaveMetrics, Inc. Lake Oswego, OR, USA) is developed to facilitate the
implementation of error-in-variables regression (including DR, WODR and YR). Two
other Igor Pro based computer programs, Histbox (Wu, 2017b) and Aethalometer data
processor (Wu, 2017c) are used for data analysis and visualization in this study.
**3  Data description**
Two types of data are used for regression comparison. The first type is synthetic data
generated by computer programs, which can be used in the EC tracer method (Turpin
and Huntzicker, 1995) to demonstrate the regression application. The true "slope" and
"intercept" are assigned during data generation, allowing quantitative comparison of
the bias of each regression scheme. The second type of data comes from ambient
measurement of light absorption, OC and EC in Guangzhou for demonstration in a real-
world application.

## 3.1 Synthetic XY data generation


In this study, numerical simulations are conducted in Igor Pro (WaveMetrics, Inc. Lake
Oswego, OR, USA) through custom codes. Two types of generation schemes are
employed, one is based on the MT pseudorandom number generator (Matsumoto and
Nishimura, 1998) and the other is based on the sine function described by Chu (2005).
The general form of linear regression on XY data can be written as:

$$Y = kX + b \tag{7}$$

Here k is the regressed slope and b is the intercept. The underlying meaning is that, Y
can be decomposed into two parts. One part is correlated with X, and the ratio is defined
by k. The other part of Y is constant and independent of X and regarded as b.
To make the discussion easier to follow, we intentionally avoid discussion using the
abstract general form and instead opt to use a real-world application case in atmospheric
science. Linear regression had been heavily applied on OC and EC data, here we use
OC and EC data as an example to demonstrate the regression application in atmospheric
science. In the EC tracer method, OC (mixture) is Y and EC (tracer) is X. OC can be
decomposed into three components based on their formation pathway:

$$OC = POC_{comb} + POC_{non-comb} + SOC \tag{8}$$

Here $POC_{comb}$ is primary OC from combustion. $POC_{non-comb}$ is primary OC emitted from
non-combustion activities. SOC is secondary OC formed during atmospheric aging.
Since $POC_{comb}$ is co-emitted with EC and well correlated with each other, their
relationship can be parameterized as:

$$POC_{comb} = (OC/EC)_{pri} \times EC \tag{9}$$

By carefully selecting an OC and EC subset when SOC is very low (considered as
approximately zero), the combination of Eqs. (8) & (9) become:

$$POC = (OC/EC)_{pri} \times EC + POC_{non-comb} \qquad (10)$$

The regressed slope of POC (Y) against EC (X) represents $(OC/EC)_{pri}$ (k in Eq. (7)). The regressed intercept become $POC_{non-comb}$ (b in Eq. (7)). With known $(OC/EC)_{pri}$ and $POC_{non-comb}$, SOC can be estimated by:

$$SOC = OC - ((OC/EC)_{pri} \times EC + POC_{non-comb}) \qquad (11)$$

The data generation starts from EC (X values). Once EC is generated, $POC_{comb}$ (the part of Y that is correlated with X) can be obtained by multiplying EC with a preset constant, $(OC/EC)_{pri}$ (slope k). Then the other preset constant $POC_{non-comb}$ is added to $POC_{comb}$ and the sum becomes POC (Y values). To simulate the real-world situation, measurement errors are added on X and Y values. Details of synthesized measurement error are discussed in the next section. Implementation of data generation by two types of mathematical schemes is explained in sect. 3.1.2 and 3.1.3, respectively.

### 3.1.1  Parameterization of synthesized measurement uncertainty

Weighting of variables is a crucial input for errors-in-variables linear regression methods such as DR, YR and WODR. In practice, the weights are usually defined as the inverse of the measurement error variance (Eq. (5)). When measurement errors are considered, measured concentrations ($Conc._{measured}$) are simulated by adding measurement uncertainties ($\varepsilon_{Conc.}$) to the true concentrations ($Conc._{true}$):

$$Conc._{measured} = Conc._{true} + \varepsilon_{Conc.} \qquad (12)$$

Here $\varepsilon_{Conc.}$ is the random error following an even distribution with an average of 0, the range of which is constrained by:

$$-\gamma_{Unc} \times Conc._{true} \leq \varepsilon_{Conc.} \leq +\gamma_{Unc} \times Conc._{true} \qquad (13)$$

The $\gamma_{Unc}$ is a dimensionless factor that describes the fractional measurement uncertainty relative to the true concentration ($Conc._{true}$). $\gamma_{Unc}$ could be a function of $Conc._{true}$ (Thompson, 1988) or a constant. The term $\gamma_{Unc} \times Conc._{true}$ defines the boundary of random measurement errors.

Two types of measurement error are considered in this study. The first type is $\gamma_{Unc-nonlinear}$. In the data generation scheme of Chu (2005) for the measurement uncertainties ($\varepsilon_{POC}$ and $\varepsilon_{EC}$), $\gamma_{Unc-nonlinear}$ is non-linearly related to $Conc._{true}$:

$$\gamma_{Unc-nonlinear} = \frac{1}{\sqrt{Conc._{true}}} \tag{14}$$

then Eq. (13) for POC and EC become:
$$-\frac{1}{\sqrt{POC_{true}}} \times POC_{true} \le \varepsilon_{POC} \le +\frac{1}{\sqrt{POC_{true}}} \times POC_{true} \tag{15}$$

$$-\frac{1}{\sqrt{EC_{true}}} \times EC_{true} \le \varepsilon_{EC} \le +\frac{1}{\sqrt{EC_{true}}} \times EC_{true} \tag{16}$$

In Eq. (14), the $\gamma_{Unc}$ decreases as concentration increases, since low concentrations are
usually more challenging to measure. As a result, the $\gamma_{Unc-nonlinear}$ defined in Eq.
(14) is more realistic than the constant approach, but there are two limitations. First, the
physical meaning of the uncertainty unit is lost. If the unit of OC is µg m$^{-3}$, then the
unit of $\varepsilon_{OC}$ becomes $\sqrt{\mu g\ m^{-3}}$. Second, the concentration is not normalized by a
consistent relative value, making it sensitive to the X and Y units used. For example, if
POC$_{true}$=0.9 µg m$^{-3}$, then $\varepsilon_{POC}$= ±0.95 µg m$^{-3}$ and $\gamma_{Unc}$ = 105%, but by changing the
concentration unit to POC$_{true}$=900 ng m$^{-3}$, then $\varepsilon_{OC}$= ±30 ng m$^{-3}$ and $\gamma_{Unc}$ = 3%. To
overcome these deficiencies, we propose to modify Eq. (14) to:
$$\gamma_{Unc} = \sqrt{\frac{LOD}{Conc._{true}}} \times \alpha \tag{17}$$

here LOD (limit of detection) is introduced to generate a dimensionless $\gamma_{Unc}$. $\alpha$ is a
dimensionless adjustable factor to control the position of $\gamma_{Unc}$ curve on the
concentration axis, which is indicated by the value of $\gamma_{Unc}$ at LOD level. As shown in
Fig. 1a, at different values of $\alpha$ ($\alpha$ =1, 0.5 and 0.3), the corresponding $\gamma_{Unc}$ at the same
LOD level would be 100%, 50% and 30%, respectively. By changing $\alpha$, the location of
the $\gamma_{Unc}$ curve on X axis direction can be set, using the $\gamma_{Unc}$ at LOD as the reference
point. Then Eq. (17) for POC and EC become:
$$-\sqrt{\frac{LOD_{POC}}{POC_{true}}} \times \alpha_{POC} \times POC_{true} \le \varepsilon_{POC} \le +\sqrt{\frac{LOD_{POC}}{POC_{true}}} \times \alpha_{POC} \times POC_{true}$$

$$\tag{18}$$

$$-\sqrt{\frac{LOD_{EC}}{EC_{true}}} \times \alpha_{EC} \times EC_{true} \le \varepsilon_{EC} \le +\sqrt{\frac{LOD_{EC}}{EC_{true}}} \times \alpha_{EC} \times EC_{true} \tag{19}$$

With the modified $\gamma_{Unc-nonlinear}$ parameterization, concentrations of POC and EC are
normalized by a corresponding LOD, which maintains unit consistency between
POC_true and $\varepsilon_{POC}$ and EC_true and $\varepsilon_{EC}$, and eliminates dependency on the concentration
unit.
Uniform distribution has been used in previous studies (Cox et al., 2003; Chu, 2005;
Saylor et al., 2006) and is adopted in this study to parameterize measurement error. For
a uniform distribution in the interval [a,b], the variance is $\frac{1}{12}(a-b)^2$. Since $\varepsilon_{POC}$ and
$\varepsilon_{EC}$ follow a uniform distribution in the interval as given by Eqs. (18) and (19), the
weights in DR and YR (inverse of variance) become:
$$\omega(X_i) = \frac{1}{\sigma_{X_i}^2} = \frac{3}{EC_{true} \times LOD_{EC} \times \alpha_{EC}^2} \tag{20}$$

$$\omega(Y_i) = \frac{1}{\sigma_{Y_i}^2} = \frac{3}{POC_{true} \times LOD_{POC} \times \alpha_{POC}^2} \tag{21}$$

The parameter $\lambda$ in Deming regression is then determined:
$$\lambda = \frac{\omega(X_i)}{\omega(Y_i)} = \frac{POC_{true} \times LOD_{POC} \times \alpha_{POC}^2}{EC_{true} \times LOD_{EC} \times \alpha_{EC}^2} \tag{22}$$

Besides the $\gamma_{Unc-nonlinear}$ discussed above, a second type measurement uncertainty
parameterized by a constant proportional factor, $\gamma_{Unc-linear}$, is very common in
atmospheric applications:
$$-\gamma_{POCunc} \times POC_{true} \leq \varepsilon_{POC} \leq +\gamma_{POCunc} \times POC_{true} \tag{23}$$

$$-\gamma_{ECunc} \times EC_{true} \leq \varepsilon_{EC} \leq +\gamma_{ECunc} \times EC_{true} \tag{24}$$

where $\gamma_{POCunc}$ and $\gamma_{ECunc}$ are the relative measurement uncertainties, e.g., for relative
measurement uncertainty of 10%, $\gamma_{Unc}$=0.1. As a result, the measurement error is
linearly proportional to the concentration. An example comparison of $\gamma_{Unc-nonlinear}$
and $\gamma_{Unc-linear}$ is shown in Fig. 1b. For $\gamma_{Unc-linear}$, the weights become:
$$\omega(X_i) = \frac{1}{\sigma_{X_i}^2} = \frac{3}{(\gamma_{ECunc} \times EC_{true})^2} \tag{25}$$

$$\omega(Y_i) = \frac{1}{\sigma_{Y_i}^2} = \frac{3}{(\gamma_{POCunc} \times POC_{true})^2} \tag{26}$$

and $\lambda$ for Deming regression can be determined:
$$\lambda = \frac{\omega(X_i)}{\omega(Y_i)} = \frac{(\gamma_{POCunc} \times POC_{true})^2}{(\gamma_{ECunc} \times EC_{true})^2} \tag{27}$$

### 3.1.2 XY data generation by Mersenne Twister (MT) generator following a specific distribution

The Mersenne twister (MT) is a pseudorandom number generator (PRNG) developed by Matsumoto and Nishimura (1998). MT has been widely adopted by mainstream numerical analysis software (e.g., Matlab, SPSS, SAS and Igor Pro) as well as popular programing languages (e.g., R, Python, IDL, C++ and PHP). Data generation using MT provides a few advantages: (1) Frequency distribution can be easily assigned during the data generation process, allowing straightforward simulation of the frequency distribution characteristics (e.g., Gaussian or Log-normal) observed in ambient measurements; (2) The inputs for data generation are simply the mean and standard deviation of the data series and can be changed easily by the user; (3) The correlation ($R^2$) between X and Y can be manipulated easily during the data generation to satisfy various purposes; (4) Unlike the sine function described by Chu (2005) that has a sample size limitation of 120, the sample size in MT data generation is highly flexible.

In this section, we will use POC as Y and EC as X as an example to explain the data generation. Procedure of applying MT to simulate ambient POC and EC data can be found in our previous study (Wu and Yu, 2016). Details of the data generation steps are shown in Fig. 2 and described below. The first step is generation of $EC_{true}$ by MT. In our previous study, it was found that ambient POC and EC data follow a lognormal distribution in various locations of the Pearl River Delta (PRD) region. Therefore, lognormal distributions are adopted during $EC_{true}$ generation. A range of average concentration and relative standard deviation (RSD) from ambient samples is considered in formulating the lognormal distribution. The second step is to generate $POC_{comb}$. As shown in Fig. 2, $POC_{comb}$ is generated by multiplying $EC_{true}$ with $(OC/EC)_{pri}$. Instead of having a Gaussian distribution, $(OC/EC)_{pri}$ in this study is a single value, which favors direct comparison between the true value of $(OC/EC)_{pri}$ and $(OC/EC)_{pri}$ estimated from the regression slope. The third step is generation of $POC_{true}$ by adding $POC_{non-comb}$ onto $POC_{comb}$. Instead of having a distribution, $POC_{non-comb}$ in this study is a single value, which favors direct comparison between the true value of $POC_{non-comb}$ and $POC_{non-comb}$ estimated from the regression intercept. The fourth step is to compute $\varepsilon_{POC}$ and $\varepsilon_{EC}$. As discussed in sect. 3.1.1, two types of measurement errors are considered for $\varepsilon_{POC}$ and $\varepsilon_{EC}$ calculation: $\gamma_{Unc-nonlinear}$ and $\gamma_{Unc-linear}$. In the

last step, POC_measured and EC_measured are calculated following Eq. (12), i.e., applying
measurement errors on POC_true and EC_true. Then POC_measured and EC_measured can be used
as Y and X, respectively, to test the performance of various regression techniques. An
Igor Pro based program with graphical user interface (GUI) is developed to facilitate
the MT data generation for OC and EC. A brief introduction is given in the
Supplemental Information.

### 3.1.3  XY data generation by the sine function of Chu (2005)

Beside MT, inclusion of the sine function data generation scheme in this study mainly
serves two purposes. First, the sine function scheme was adopted in two previous
studies (Chu, 2005; Saylor et al., 2006), the inclusion of this scheme can help to verify
whether the codes in Igor for various regression approaches yield the same results from
the two previous studies. Second, the crosscheck between results from sine function
and MT provides circumstantial evidence that the MT scheme works as expected.
In this section, XY data generation by sine functions is demonstrated using POC as Y
and EC as X. There are four steps in POC and EC data generation as shown by the
flowchart in Fig. S2. Details are explained as follows: (1) The first step is to generate
POC and EC (Chu, 2005):

$$POC_{comb} = 14 + 12(\sin(\frac{x}{\tau}) + \sin(x - \phi)) \tag{28}$$

$$EC_{true} = 3.5 + 3(\sin(\frac{x}{\tau}) + \sin(x - \phi)) \tag{29}$$

Here x is the elapsed hour (x=1,2,3……n; n≤120), $\tau$ is used to adjust the width of each
peak, and $\phi$ is used to adjust the phase of the sine wave. The constants 14 and 3.5 are
used to lift the sine wave to the positive range of the Y axis. An example of data
generation by the sine functions of Chu (2005) is shown in Fig. 3. Dividing Eq. (28) by
Eq. (29) yields a value of 4. In this way the exact relation between POC and EC is
defined clearly as $(OC/EC)_{pri}$ = 4. (2) With POC_comb and EC_true generated, the second
step is to add POC_non-comb to POC_comb to compute POC_true. As for POC_non-comb, a single
value is assigned and added to all POC following Eq. (10). Then the goodness of the
regression intercept can be evaluated by comparing the regressed intercept with preset
POC_non-comb. (3) The third step is to compute $\varepsilon_{POC}$ and $\varepsilon_{EC}$ , considering both
$\gamma_{Unc-nonlinear}$ and $\gamma_{Unc-linear}$. (4) The last step is to apply measurement errors on
POC$_{true}$ and EC$_{true}$ following Eq. (12). Then POC$_{measured}$ and EC$_{measured}$ can be used as
Y and X, respectively, to evaluate the performance of various regression techniques.

## 3.2    Ambient measurement of σ$_{abs}$ and EC

Sampling was conducted from Feb 2012 to Jan 2013 at the suburban Nancun (NC) site
(23° 0'11.82"N, 113°21'18.04"E), which is situated on top of the highest peak (141 m
ASL) in the Panyu district of Guangzhou. This site is located at the geographic center
of Pearl River Delta region (PRD), making it a good location for representing the
average atmospheric mixing characteristics of city clusters in the PRD region. Light
absorption measurements were performed by a 7$\lambda$ Aethalometer (AE-31, Magee
Scientific Company, Berkeley, CA, USA). EC mass concentrations were measured by
a real time ECOC analyzer (Model RT-4, Sunset Laboratory Inc., Tigard, Oregon,
USA). Both instruments utilized inlets with a 2.5 μm particle diameter cutoff. The algorithm
of Weingartner et al. (2003) was adopted to correct the sampling artifacts (aerosol
loading, filter matrix and scattering effect) (Coen et al., 2010) in Aethalometer
measurement. A customized computer program with graphical user interface,
Aethalometer data processor (Wu et al., 2018), was developed to perform the data
correction and detailed descriptions can be found in
https://sites.google.com/site/wuchengust. More details of the measurements can be
found in Wu et al. (2018).

## 4    Comparison study using synthetic data

In the following comparisons, six regression approaches are compared using two data
generation schemes (Chu sine function and MT) separately, as illustrated in Fig. 4. Each
data generation scheme considers both $\gamma_{Unc-nonlinear}$ and $\gamma_{Unc-linear}$ in measurement
error parameterization. In total, 18 cases are tested with different combination of data
generation schemes, measurement error parameterization schemes, true slope and
intercept settings. In each case, six regression approaches are tested, including OLS,
DR ($\lambda = 1$), DR ($\lambda = \frac{\omega(X_i)}{\omega(Y_i)}$), ODR, WODR and YR. In commercial software (e.g.,
OriginPro®, SigmaPlot®, GraphPad Prism®, etc), $\lambda$ in DR is set to 1 by default if not
specified. As indicated by Saylor et al. (2006), the bias observed in the study of Chu
(2005) is likely due to $\lambda = 1$ in DR. The purpose of including DR ($\lambda = 1$) in this study
is to examine the potential bias using the default input in many software products. The
six regression approaches are considered to examine the sensitivity of regression results
to various parameters used in data generation. For each case, 5000 runs are performed
to obtain statistically significant results, as recommended by Saylor et al. (2006). The
mean slope and intercept from 5000 runs is compared with the true value assigned
during data generation. If the difference is <5%, the result is considered unbiased.

### 368    4.1    Comparison results using the data set of Chu (2005)

In this section, the scheme of Chu (2005) is adopted for data generation to obtain a
benchmark of six regression approaches. With different setup of slope, intercept and
$\gamma_{Unc}$ , 6 cases (Case 1 ~ 6) are studied and the results are discussed below.

### 372    4.1.1    Results with $\gamma_{Unc-nonlinear}$

A comparison of the regression techniques results with $\gamma_{Unc-nonlinear}$ (following Eqs.
(18) & (19)) is summarized in Table 2. $LOD_{POC}$ , $LOD_{EC}$, $\alpha_{POC}$ and $\alpha_{EC}$ are all set to 1
to reproduce the data studied by Chu (2005) and Saylor et al. (2006). Two sets of true
slope and intercept are considered (Case 1: Slope=4, Intercept=0; Case 2: Slope=4,
Intercept=3) to examine if any results are sensitive to the non-zero intercept. The $R^2$
(POC, EC) from 5000 runs for both case 1 and 2 are 0.67±0.03.
As shown in Fig. 5, for the zero-intercept case (Case 1), OLS significantly
underestimates the slope (2.95±0.14) while overestimates the intercept (5.84±0.78).
This result indicates that OLS is not suitable for errors-in-variables linear regression,
consistent with similar analysis results from Chu (2005) and Saylor et al. (2006). With
DR, if the λ is properly calculated by weights ($\lambda = \frac{\omega(X_i)}{\omega(Y_i)}$), unbiased slope (4.01±0.25)
and intercept (-0.04±1.28) are obtained; however, results from DR with λ=1 show
obvious bias in the slope (4.27±0.27) and intercept (-1.45±1.36). ODR also produces
biased slope (4.27±0.27) and intercept (-1.45±1.36), which are identical to results of
DR when λ=1. With WODR, unbiased slope (3.98±0.22) is observed, but the intercept
is overestimated (1.12±1.02). Results of YR are identical to WODR. For Case 2
(slope=4, intercept=3), slopes from all six regression approaches are consistent with
Case 1 (Table 2). The Case 2 intercepts are equal to the Case 1 intercepts plus 3,
implying that all the regression methods are not sensitive to a non-zero intercept.
For case 3, $LOD_{POC}$=0.5, $LOD_{EC}$=0.5, $\alpha_{POC}$=0.5, $\alpha_{EC}$=0.5 are adopted (Table 2),
leading to an offset to the left of $\gamma_{Unc-nonlinear}$ (blue curve) compared to Case 1 and 2
(black curve) in Fig. 1. As a result, for the same concentration of EC and OC in Case
3, the $\gamma_{Unc-nonlinear}$ is smaller than in Case 1 and Case 2 as indicated by a higher $R^2$
(0.95±0.01 for Case 3, Table 2). With a smaller measurement uncertainty, the degree
of bias in Case 3 is smaller than in Case 1. For example, OLS slope is less biased in
Case 3 (3.83±0.08) compared to Case 1 (2.94±0.14). Similarly, the slope (4.03±0.09)
and intercept (-0.18±0.44) of DR ($\lambda$=1) exhibit a much smaller bias with a smaller
measurement uncertainty, implying that the degree of bias by improperly weighting in
DR, WODR and YR is associated with the degree of measurement uncertainty. A higher
measurement uncertainty results in larger bias in slope and intercept.
An uneven $LOD_{POC}$ and $LOD_{EC}$ is tested in Case 4 with $LOD_{POC}$=1, $LOD_{EC}$=0.5,
$\alpha_{POC}$=0.5, $\alpha_{EC}$=0.5, which yield a $R^2$(POC, EC) of 0.78±0.02. The results are similar
to Case 1. For DR ($\lambda = \frac{\omega(X_i)}{\omega(Y_i)}$) unbiased slope and intercept are obtained. For WODR
and YR, unbiased slopes are reported with a small bias in the intercepts. Large bias
values are observed in both the slopes and intercepts in Case 4 using OLS, DR ($\lambda = 1$)
and ODR.

### 409    **4.1.2 Results with $\gamma_{Unc-linear}$**

Cases 5 and 6 represent the results from using $\gamma_{Unc-linear}$ and are shown in Table 2.
$\gamma_{Unc}$ is set to 30% to achieve a $R^2$ (POC, EC) of 0.7, a value close to the $R^2$ in studies
of Chu (2005) and Saylor et al. (2006). In Case 5 (slope=4, intercept=0), unbiased
slopes and intercepts are determined by DR ($\lambda = \frac{\omega(X_i)}{\omega(Y_i)}$), WODR and YR. OLS
underestimates the slope (3.32 ±0.20) and overestimates intercept (3.77 ±0.90), while
DR ($\lambda = 1$) and ODR overestimate the slopes (4.75 ±0.30) and underestimate the
intercepts (-4.14 ±1.36). In Case 6 (slope=4, intercept=3), results similar to Case 5 are
obtained. It is worth noting that although the mean intercept (3.05±1.22) of DR ($\lambda =$
$\frac{\omega(X_i)}{\omega(Y_i)}$), is closest to the true value (intercept=3), the deviations are much larger than for
WODR (2.72±0.74).

## 4.2   Comparison results using data generated by MT

In this section, MT is adopted for data generation to obtain a benchmark of six
regression approaches. Both $\gamma_{Unc-nonlinear}$ and $\gamma_{Unc-linear}$ are considered. With
different configuration of slope, intercept and $\gamma_{Unc}$, 12 cases (Case 7 ~ Case 18) are
studied and the results are discussed below.

### 4.2.1   $\gamma_{Unc-nonlinear}$ results

Cases 7 and 8 use data generated by MT and $\gamma_{Unc-nonlinear}$ with results shown in Table
2. In Case 7 (slope=4, intercept=0, $LOD_{POC}$=1, $LOD_{EC}$=1, $\alpha_{POC}$=1, $\alpha_{EC}$=1), unbiased
slope (4.00 ±0.03) and intercept (0.00 ±0.17) is estimated by DR ($\lambda = \frac{\omega(X_i)}{\omega(Y_i)}$). WODR
and YR yield unbiased slopes (3.96 ±0.03) but overestimate the intercepts (1.21 ±0.13).
DR ($\lambda = 1$) and ODR report slightly biased slopes (4.17 ±0.04) with biased intercepts
(-0.94 ±0.18). OLS underestimates the slope (3.22 ±0.03) and overestimates the
intercept (4.30 ±0.14). In Case 8 (slope=4, intercept=3, $LOD_{POC}$=1, $LOD_{EC}$=1, $\alpha_{POC}$=1,
$\alpha_{EC}$=1), DR ($\lambda = \frac{\omega(X_i)}{\omega(Y_i)}$) provides unbiased slope (4.00 ±0.03) and intercept (3.00 ±0.18)
estimations. WODR and YR report unbiased slopes (3.97 ±0.03) and overestimate
intercepts (4.11 ±0.13). OLS, DR ($\lambda = 1$) and ODR report biased slopes and intercepts.
To test the overestimation/underestimation dependency on the true slope, Case 9
(slope=0.5, intercept=0, $LOD_{POC}$ =1, $LOD_{EC}$ =1, $\alpha_{POC}$ =1, $\alpha_{EC}$ =1) and case 10
(slope=0.5, intercept=3, $LOD_{POC}$=1, $LOD_{EC}$=1, $\alpha_{POC}$=1, $\alpha_{EC}$=1) are conducted and the
results are shown in Table 2. Unlike the overestimation observed in Case 1~Case 8, DR
($\lambda = 1$) and ODR underestimate the slopes (0.46 ±0.01) in Case 9. In case 10, DR ($\lambda =$
1), DR ($\lambda = \frac{\omega(X_i)}{\omega(Y_i)}$) and ODR report unbiased slopes and intercepts. Case 11 and case
12 test the bias when the true slope is 1 as shown in Table 2. In Case 11 (intercept=0),
all regression approaches except OLS can provide unbiased results. In Case 12, all
regression approaches report unbiased slopes except OLS, but DR ($\lambda = \frac{\omega(X_i)}{\omega(Y_i)}$) is the
only regression approach that reports unbiased intercept.
These results imply that if the true slope is less than 1, the improper weighting ($\lambda = 1$)
in Deming regression and ODR without weighting tends to underestimate slope. If the
true slope is 1, these two estimators can provide unbiased results. If the true slope is
larger than 1, the improper weighting ($\lambda = 1$) in Deming regression and ODR without
weighting tends to overestimate slope.
**4.2.2 $\gamma_{Unc-linear}$ results**
Cases 13 and 14 (Table 2) represent the results from using $\gamma_{Unc-linear}$ (30%) and data
generated from MT. For case 13 (slope=4, intercept=0), DR ($\lambda = \frac{\omega(X_i)}{\omega(Y_i)}$), WODR and
YR provide the best estimation of slopes and intercepts. DR ($\lambda = 1$) and ODR
overestimate slopes (4.53 ±0.05) and underestimate intercepts (-2.94 ±0.24). For case
14 (slope=4, intercept=3), DR ($\lambda = \frac{\omega(X_i)}{\omega(Y_i)}$), WODR and YR provide an unbiased
estimation of slopes. But DR ($\lambda = \frac{\omega(X_i)}{\omega(Y_i)}$) is the only regression approach reporting
unbiased intercept (3.08 ±0.23). Cases 15 and 16 are tested to investigate whether the
results are different if the true slope is smaller than 1. As shown in Table 2, the results
are similar to case 13&14 that DR ($\lambda = \frac{\omega(X_i)}{\omega(Y_i)}$) can provide unbiased slope and intercept
while WODR and YR can provide unbiased slopes but biased intercepts. Cases 17 and
18 are tested to see if the results are the same for a special case when the true slope is
1. As shown in Table 2, the results are similar to case 13&14, implying that these results
are not sensitive to the special case when the true slope is 1.
**4.3 The importance of appropriate λ input for Deming regression**
As discussed above, inappropriate λ assignment in the Deming regression (e.g., λ=1 by
default for much commercial software) leads to biased slope and intercept. Beside λ=1,
inappropriate λ input due to improper handling of measurement uncertainty can also
result in bias for Deming regression. An example is shown in Fig. S3. Data is generated
by MT with following parameters: slope=4, intercept=0, and $\gamma_{Unc-linear}$ (30%). Fig.
S2 a&b demonstrates that when an appropriate λ is provided (following $\gamma_{Unc-linear}$,
$\lambda = \frac{POC^2}{EC^2}$), unbiased slopes and intercepts are obtained. If an improper λ is used due to
a mismatched measurement uncertainty assumption ($\gamma_{Unc-nonlinear}$, $\lambda = \frac{POC}{EC}$), the
slopes are overestimated (Fig. S3c, 4.37±0.05) and intercepts are underestimated (Fig.
S3d,-2.01±0.24). This result emphasizes the importance of determining the correct
form of measurement uncertainty in ambient samples, since $\lambda$ is a crucial parameter in
Deming regression.
In the $\lambda$ calculation, different representations for POC and EC, including mean, median
and mode, are tested as shown in Fig. S4. The results show that when X and Y have a
similar distribution (e.g., both are log-normal), any of mean, median or mode can be
used for the $\lambda$ calculation.

## 482 **4.4 Caveats of regressions with unknown X and Y uncertainties**

In atmospheric applications, there are scenarios in which a priori error in one of the
variables is unknown, or the measurement error described cannot be trusted. For
example, in the case of comparing model prediction and measurement data, the
uncertainty of model prediction data is unknown. A second example is the case in which
measurement uncertainty cannot be determined due to the lack of duplicated or
collocated measurements and as a result, an arbitrarily assumed uncertainty is used.
Such a case was illustrated in the study by Flanagan et al. (2006). They found that in the
Speciation Trends Network (STN), the whole-system uncertainty retrieved by data from
collocated samplers was different from the arbitrarily assumed 5% uncertainty.
Additionally, the discrepancy between the actual uncertainty obtained through
collocated samplers and the arbitrarily assumed uncertainty varied by chemical species.
To investigate the performance of different regression approaches in these cases, two
tests (A and B) are conducted.
In Test A, the actual measurement error for X is fixed at 30% while $\gamma_{Unc}$ for Y varies
from 1% to 50%. The assumed measurement error for regression is 10% for both X and
Y. Result of Test A are shown in Figs. 6 a and b. For OLS, the slopes are under-
estimated (-14 $\sim$ -12%) and intercepts are overestimated (90 $\sim$ 103%) and the biases are
independent of variations in $\gamma_{Unc\_Y}$. ODR and DR ($\lambda = 1$) yield similar results with
over-estimated slopes (0 $\sim$ 44%) and under-estimated intercepts (-330 $\sim$ 0%). The degree
of bias in slopes and intercepts depends on the $\gamma_{Unc\_Y}$. WODR, DR ($\lambda = \frac{\omega(X_i)}{\omega(Y_i)}$) and YR
perform much better than other regression approaches in Test A, with a smaller bias in
both slopes (-8 ~ 12%) and intercepts -98 ~ 55%).
In Test B, $\gamma_{Unc\_Y}$ is fixed at 30% and $\gamma_{Unc\_X}$ varies between 1 ~ 50%. The results of Test
B are shown in Figs. 6 c and d. The assumed measurement error for regression is 10%
for both X and Y. OLS underestimates the slopes (-29 ~-0.2%) and overestimates the
intercepts (2 ~ 209%). In contrast to Test A in which slope and intercept biases are
independent of variations in $\gamma_{Unc\_Y}$, the slope and intercept biases in Test B exhibit
dependency on $\gamma_{Unc\_X}$. The reason behind is because OLS only considers errors in Y
and X is assumed to be error free. ODR and DR ($\lambda = 1$) yield similar results with over-
estimated slopes (11 ~ 18%) and under-estimated intercepts ( -144 ~ -87%). The degree
of bias in slopes and intercepts is relatively independent on the $\gamma_{Unc\_X}$. WODR, DR
($\lambda = \frac{\omega(X_i)}{\omega(Y_i)}$) and YR performed much better than the other regression approaches in Test
B, with a smaller bias in both slopes (-14 ~ 8%) and intercepts (-59 ~ 106%).
The results from these two tests suggest that, in case of one of the measurement error
described cannot be trusted or a priori error in one of the variables is unknown, WODR,
DR ($\lambda = \frac{\omega(X_i)}{\omega(Y_i)}$) and YR should be used instead of ODR and DR ($\lambda = 1$) and OLS. This
conclusion is consistent with results presented in sect. 4.1 and 4.2. This analysis, albeit
crude, also suggests that, in general, the magnitude of bias in slope estimation by these
regression approaches is smaller than those for intercept. In other words, slope is a more
reliable quantity compared to intercept when extracting quantitative information from
linear regressions.
**5   Regression applications to ambient data**
This section demonstrates the application of the 6 regression approaches on a light
absorption coefficient and EC dataset collected in a suburban site in Guangzhou. As
mentioned in sect. 4.4, measurement uncertainties are crucial inputs for DR, YR and
WODR. The measurement precision of Aethalometer is 5% (Hansen, 2005) while EC
by RT-ECOC analyzer is 24% (Bauer et al., 2009). These measurement uncertainties
are used in DR, YR and WODR calculation. The data-set contains 6926 data points
with a $R^2$ of 0.92.
As shown in Fig. 7, Y axis is light absorption at 520 nm ($\sigma_{abs520}$) and the X axis is EC
mass concentration. The regressed slopes represent the mass absorption efficiency
(MAE) of EC at 520 nm, ranging from 13.66 to 15.94 $m^2g^{-1}$ by the six regression
approaches. OLS yields the lowest slope (13.66 as shown in Fig. 7a) among all six
regression approaches, consistent with the results using synthetic data. This implies that
OLS tends to underestimate regression slope when mean Y to X ratio is larger than 1.
DR ($\lambda = 1$) and ODR report the same slope (14.88) and intercept (5.54), this
equivalency is also observed for the synthetic data. Similarly, WODR and YR yield
identical slope (14.88) and intercept (5.54), in line with the synthetic data results. The
regressed slope by DR ($\lambda = 1$) is higher than DR ($\lambda = \frac{\omega(X_i)}{\omega(Y_i)}$), and this relationship
agrees well with the synthetic data results.
Regression comparison is also performed on hourly OC and EC data. Regression on
OC/EC percentile subset is a widely used empirical approach for primary OC/EC ratio
determination. Fig. S5 shows the regression slopes as a function of OC/EC percentile.
OC/EC percentile ranges from 0.5% to 100%, with an interval of 0.5%. As the
percentile increases, SOC contribution in OC increases as well, resulting in decreased
$R^2$ between OC and EC. The deviations between six regression approaches exhibit a
dependency on $R^2$. When percentile is relatively small (e.g., <10%), the differences
between the six regression approaches are also small due to the high $R^2$ (0.98). The
deviations between the six regression approaches become more pronounced as $R^2$
decreases (e.g., <0.9). The deviations are expected to be even larger when $R^2$ is less
than 0.8. These results emphasize the importance of applying error-in-variables
regression, since ambient XY data more likely has a $R^2$ less than 0.9 in most cases.
As discussed in this section, the ambient data confirm the results obtained in comparing
methods with the synthetic data. The advantage of using the synthetic data for
regression approaches evaluation is that the ideal slope and intercept are known values
during the data generation, so the bias of each regression approach can be quantified.
**6    Recommendations and conclusions**
This study aims to provide a benchmark of commonly used linear regression algorithms
using a new data generation scheme (MT). Six regression approaches are tested,
including OLS, DR ($\lambda = 1$), DR ($\lambda = \frac{\omega(X_i)}{\omega(Y_i)}$), ODR, WODR and YR. The results show
that OLS fails to estimate the correct slope and intercept when both X and Y have
measurement errors. This result is consistent with previous studies. For ambient data
with $R^2$ less than 0.9, error-in-variables regression is needed to minimize the biases in
slope and intercept. If measurement uncertainties in X and Y are determined during the
measurement, measurement uncertainties should be used for regression. With
appropriate weighting, DR, WODR and YR can provide the best results among all
tested regression techniques. Sensitivity tests also reveal the importance of the
weighting parameter λ in DR. An improper λ could lead to biased slope and intercept.
Since the λ estimation depends on the form of the measurement errors, it is important
to determine the measurement errors during the experimentation stage rather than
making assumptions. If measurement errors are not available from the measurement
and assumptions are made on measurement errors, DR, WODR and YR are still the
best option that can provide the least bias in slope and intercept among all tested
regression techniques. For these reasons, DR, WODR and YR are recommended for
atmospheric studies when both X and Y data have measurement errors.
Application of error-in-variables regression is often overlooked in atmospheric studies,
partly due to the lack of a specified tool for the regression implementation. To facilitate
the implementation of error-in-variables regression (including DR, WODR and YR), a
computer program (Scatter plot) with graphical user interface (GUI) in Igor Pro
(WaveMetrics, Inc. Lake Oswego, OR, USA) is developed (Fig. 8). It is packed with
many useful features for data analysis and plotting, including batch plotting, data
masking via GUI, color coding in Z axis, data filtering and grouping by numerical
values and strings. The Scatter plot program and user manual are available from
https://sites.google.com/site/wuchengust and https://doi.org/10.5281/zenodo.832417.

**Appendix A: Equations of regression techniques**
Ordinary Least Square (**OLS**) calculation steps.
First calculate average of observed $X_i$ and $Y_i$.
$$\bar{X} = \frac{\sum_{i=1}^{N} X_i}{N} \tag{A1}$$

$$\bar{Y} = \frac{\sum_{i=1}^{N} Y_i}{N} \tag{A2}$$

Then calculate $S_{xx}$ and $S_{yy}$.
$$S_{xx} = \sum_{i=1}^{N}(X_i - \bar{X})^2 \tag{A3}$$

$$S_{yy} = \sum_{i=1}^{N}(Y_i - \bar{Y})^2 \tag{A4}$$

OLS slope and intercept can be obtained from,
$$k = \frac{S_{yy}}{S_{xx}} \tag{A6}$$

$$b = \bar{Y} - k\bar{X} \tag{A7}$$


Deming regression (**DR**) calculation steps (York, 1966).
Besides $S_{xx}$ and $S_{yy}$ as shown above, $S_{xy}$ can be calculated from,
$$S_{xy} = \sum_{i=1}^{N}(X_i - \bar{X})(Y_i - \bar{Y}) \tag{A8}$$

DR slope and intercept can be obtained from,
$$k = \frac{S_{yy} - \lambda S_{xx} + \sqrt{(S_{yy} - \lambda S_{xx})^2 + 4\lambda S_{xy}^2}}{2S_{xy}} \tag{A9}$$

$$b = \bar{Y} - k\bar{X} \tag{A10}$$


York regression (**YR**) iteration steps (York et al., 2004).
Slope by OLS can be used as the initial k in $W_i$ calculation.
$$W_i = \frac{\omega(X_i)\omega(Y_i)}{\omega(X_i) + k^2\omega(Y_i) - 2kr_i\sqrt{\omega(X_i)\omega(Y_i)}} \tag{A11}$$

$$U_i = X_i - \bar{X} = X_i - \frac{\sum_{i=1}^{N} W_i X_i}{\sum_{i=1}^{N} W_i} \tag{A12}$$

$$V_i = Y_i - \bar{Y} = Y_i - \frac{\sum_{i=1}^{N} W_i Y_i}{\sum_{i=1}^{N} W_i} \tag{A13}$$

Then calculate $\beta_i$.

$$\beta_i = W_i \left[ \frac{U_i}{\omega(Y_i)} + \frac{k V_i}{\omega(X_i)} - [k U_i + V_i] \frac{r_i}{\sqrt{\omega(X_i)\omega(Y_i)}} \right] \tag{A14}$$

Slope and intercept can be obtained from,

$$k = \frac{\sum_{i=1}^{N} W_i \beta_i V_i}{\sum_{i=1}^{N} W_i \beta_i U_i} \tag{A15}$$

$$b = \bar{Y} - k \bar{X} \tag{A16}$$

Since $W_i$ and $\beta_i$ are functions of k, k must be solved iteratively by repeating A11 to A15. If the difference between the k obtained from A15 and the k used in A11 satisfies the predefined tolerance ($\frac{k_{i+1} - k_i}{k_i} < e^{-15}$), the calculation is considered as converged. The calculation is straightforward and usually converged in 10 iterations. For example, the iteration count on the data set of Chu (2005) is around 6.

*Data availability*. OC, EC and $\sigma_{abs}$ data used in this study are available from the corresponding authors upon request. The computer programs used for data analysis and visualization in this study are available in Wu (2017a–c).

*Competing interests*. The authors declare that they have no conflict of interest.

**Acknowledgements**

This work is supported by the National Natural Science Foundation of China (Grant No. 41605002, 41475004 and 21607056), NSFC of Guangdong Province (Grant No. 2015A030313339), Guangdong Province Public Interest Research and Capacity Building Special Fund (Grant No. 2014B020216005). The author would like to thank Dr. Bin Yu Kuang at HKUST for discussion on mathematics and Dr. Stephen M Griffith at HKUST for valuable comments.

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

**Table 1.** Summary of abbreviations and symbols.

| Abbreviation/symbol | Definition |
|---|---|
| $\alpha$ | a dimensionless adjustable factor to control the position of $\gamma_{Unc}$ curve on the concentration axis |
| $b$ | intercept in linear regression |
| $\beta_i, U_i, V_i, W_i$ | intermediates in York regression calculations |
| $\gamma_{Unc}$ | fractional measurement uncertainties relative to the true concentration (%) |
| DR | Deming regression |
| $\varepsilon_{EC}, \varepsilon_{POC}$ | absolute measurement uncertainties of EC and POC |
| EC | elemental carbon |
| $EC_{true}$ | numerically synthesized true EC concentration without measurement uncertainty |
| $EC_{measured}$ | EC with measurement error ($EC_{true} + \varepsilon_{EC}$) |
| $\lambda$ | $\omega(X_i)$ to $\omega(Y_i)$ ratio in Deming regression |
| $k$ | slope in linear regression |
| LOD | limit of detection |
| MT | Mersenne twister pseudorandom number generator |
| OC | organic carbon |
| OC/EC | OC to EC ratio |
| $(OC/EC)_{pri}$ | primary OC/EC ratio |
| $OC_{non-comb}$ | OC from non-combustion sources |
| ODR | orthogonal distance regression |
| OLS | ordinary least squares regression |
| POC | primary organic carbon |
| $POC_{comb}$ | numerically synthesized true POC from combustion sources (well correlated with $EC_{true}$), measurement uncertainty not considered |
| $POC_{non-comb}$ | numerically synthesized true POC from non-combustion sources (independent of $EC_{true}$) without considering measurement uncertainty |
| $POC_{true}$ | sum of $POC_{comb}$ and $POC_{non-comb}$ without considering measurement uncertainty |
| $POC_{measured}$ | POC with measurement error ($POC_{true} + \varepsilon_{POC}$) |
| $\sigma_{X_i}, \sigma_{Y_i}$ | the standard deviation of the error in measurement of $X_i$ and $Y_i$ |
| $r_i$ | correlation coefficient between errors in $X_i$ and $Y_i$ in YR |
| S | sum of squared residuals |
| SOC | secondary organic carbon |
| $\tau$ | parameter in the sine function of Chu (2005) that adjusts the width of each peak |
| $\phi$ | parameter in the sine function of Chu (2005) that adjusts the phase of the curve |
| WODR | weighted orthogonal distance regression |
| $\bar{X}, \bar{Y}$ | average of $X_i$ and $Y_i$ |
| YR | York regression |
| $\omega(X_i), \omega(Y_i)$ | inverse of $\sigma_{X_i}$ and $\sigma_{Y_i}$, used as weights in DR calculation. |


**Table 2.** Summary of six regression approaches comparison with 5000 runs for 18 cases.

| | Data generation | | | | | Results by different regression approaches | | | | | | | | | | | |
|---|---|---|---|---|---|---|---|---|---|---|---|---|---|---|---|---|---|
| | | | | | | OLS | | DR $\lambda=1$ | | DR $\lambda=\frac{\omega(X_i)}{\omega(Y_i)}$ | | ODR | | WODR | | YR | |
| Case | Data scheme | True Slope | True Intercept | $R^2$ (X, Y) | Measurement error | Slope | Intercept | Slope | Intercept | Slope | Intercept | Slope | Intercept | Slope | Intercept | Slope | Intercept |
| 1 | | 4 | 0 | 0.67±0.03 | $LOD_{POC}$=1, $LOD_{EC}$=1 | 2.94±0.14 | 5.84±0.78 | 4.27±0.27 | -1.45±1.36 | 4.01±0.25 | -0.04±1.28 | 4.27±0.27 | -1.45±1.36 | 3.98±0.22 | 1.12±1.02 | 3.98±0.22 | 1.12±1.02 |
| 2 | | 4 | 3 | 0.67±0.04 | $a_{POC}$=1, $a_{EC}$=1. | 2.95±0.15 | 8.83±0.80 | 4.32±0.28 | 1.28±1.43 | 4.01±0.26 | 2.94±1.34 | 4.32±0.28 | 1.28±1.43 | 3.99±0.23 | 3.98±1.05 | 3.99±0.23 | 3.98±1.05 |
| 3 | | 4 | 0 | 0.95±0.01 | $LOD_{POC}$=0.5, $LOD_{EC}$=0.5 $\alpha_{POC}$=0.5, $\alpha_{EC}$=0.5 | 3.83±0.08 | 0.95±0.40 | 4.03±0.09 | -0.18±0.44 | 4±0.09 | 0±0.44 | 4.03±0.09 | -0.18±0.44 | 4±0.08 | 0.12±0.37 | 4±0.08 | 0.12±0.37 |
| 4 | Chu | 4 | 0 | 0.78±0.02 | $LOD_{POC}$=1, $LOD_{EC}$=0.5 $\alpha_{POC}$=1, $\alpha_{EC}$=1 | 3.39±0.15 | 3.34±0.75 | 4.3±0.21 | -1.66±1.06 | 4±0.19 | -0.03±0.99 | 4.3±0.21 | -1.66±1.06 | 4±0.17 | 0.33±0.81 | 4±0.17 | 0.33±0.81 |
| 5 | | 4 | 0 | 0.69±0.04 | $\gamma_{Unc}$=30% | 3.32±0.20 | 3.77±0.90 | 4.75±0.30 | -4.14±1.36 | 4.01±0.25 | -0.04±1.13 | 4.75±0.30 | -4.14±1.36 | 4±0.18 | -0.01±0.59 | 4±0.18 | -0.01±0.59 |
| 6 | | 4 | 3 | 0.66±0.04 | | 3.31±0.22 | 6.79±1.02 | 4.95±0.31 | -2.26±1.48 | 3.99±0.26 | 3.05±1.22 | 4.95±0.31 | -2.26±1.48 | 4.01±0.20 | 2.72±0.74 | 4.01±0.20 | 2.72±0.74 |
| 7 | | 4 | 0 | 0.76±0.01 | | 3.22±0.03 | 4.3±0.14 | 4.17±0.04 | -0.94±0.18 | 4±0.03 | 0±0.17 | 4.17±0.04 | -0.94±0.18 | 3.96±0.03 | 1.21±0.13 | 3.96±0.03 | 1.21±0.13 |
| 8 | | 4 | 3 | 0.75±0.01 | | 3.22±0.03 | 7.29±0.14 | 4.2±0.04 | 1.88±0.18 | 4±0.03 | 3±0.18 | 4.2±0.04 | 1.88±0.18 | 3.97±0.03 | 4.11±0.13 | 3.97±0.03 | 4.11±0.13 |
| 9 | | 0.5 | 0 | 0.76±0.01 | $LOD_{POC}$=1, $LOD_{EC}$=1 | 0.43±0.00 | 0.36±0.02 | 0.46±0.01 | 0.23±0.03 | 0.5±0.01 | 0±0.03 | 0.46±0.01 | 0.23±0.03 | 0.5±0.00 | 0±0.01 | 0.5±0.00 | 0±0.01 |
| 10 | | 0.5 | 3 | 0.56±0.01 | $a_{POC}$=1, $a_{EC}$=1 | 0.43±0.01 | 3.36±0.03 | 0.5±0.01 | 3.02±0.04 | 0.49±0.01 | 3.05±0.04 | 0.5±0.01 | 3.02±0.04 | 0.51±0.01 | 2.73±0.03 | 0.51±0.01 | 2.73±0.03 |
| 11 | | 1 | 0 | 0.76±0.01 | | 0.87±0.01 | 0.72±0.05 | 1±0.01 | 0±0.06 | 1±0.01 | 0±0.06 | 1±0.01 | 0±0.06 | 1±0.01 | 0±0.02 | 1±0.01 | 0±0.02 |
| 12 | MT | 1 | 3 | 0.66±0.01 | | 0.87±0.01 | 3.72±0.05 | 1.09±0.01 | 2.52±0.07 | 0.99±0.01 | 3.07±0.06 | 1.09±0.01 | 2.52±0.07 | 1.01±0.01 | 2.71±0.04 | 1.01±0.01 | 2.7±0.04 |
| 13 | | 4 | 0 | 0.76±0.01 | | 3.48±0.04 | 2.87±0.18 | 4.53±0.05 | -2.94±0.24 | 4±0.05 | 0±0.22 | 4.53±0.05 | -2.94±0.24 | 4±0.03 | 0±0.09 | 4±0.03 | 0±0.09 |
| 14 | | 4 | 3 | 0.73±0.01 | | 3.48±0.04 | 5.87±0.19 | 4.67±0.05 | -0.67±0.26 | 3.98±0.05 | 3.08±0.23 | 4.67±0.05 | -0.67±0.26 | 4.02±0.03 | 2.68±0.11 | 4.02±0.03 | 2.68±0.11 |
| 15 | | 0.5 | 0 | 0.54±0.01 | $\gamma_{Unc}$=30% | 0.4±0.01 | 0.55±0.03 | 0.45±0.01 | 0.26±0.03 | 0.5±0.01 | 0.01±0.03 | 0.45±0.01 | 0.26±0.03 | 0.52±0.01 | -0.23±0.02 | 0.52±0.01 | -0.23±0.02 |
| 16 | | 0.5 | 3 | 0.40±0.01 | | 0.4±0.01 | 3.54±0.04 | 0.5±0.01 | 2.98±0.04 | 0.5±0.01 | 3±0.04 | 0.5±0.01 | 2.98±0.04 | 0.52±0.01 | 2.65±0.04 | 0.52±0.01 | 2.65±0.04 |
| 17 | | 1 | 0 | 0.65±0.01 | | 0.8±0.01 | 1.07±0.04 | 1±0.01 | 0±0.05 | 1±0.01 | 0±0.05 | 1±0.01 | 0±0.05 | 1±0.01 | 0±0.04 | 1±0.01 | 0±0.04 |
| 18 | | 1 | 3 | 0.59±0.01 | | 0.8±0.01 | 4.07±0.05 | 1.07±0.01 | 2.62±0.07 | 1±0.01 | 3±0.06 | 1.07±0.01 | 2.62±0.07 | 1.02±0.01 | 2.84±0.05 | 1.02±0.01 | 2.84±0.05 |


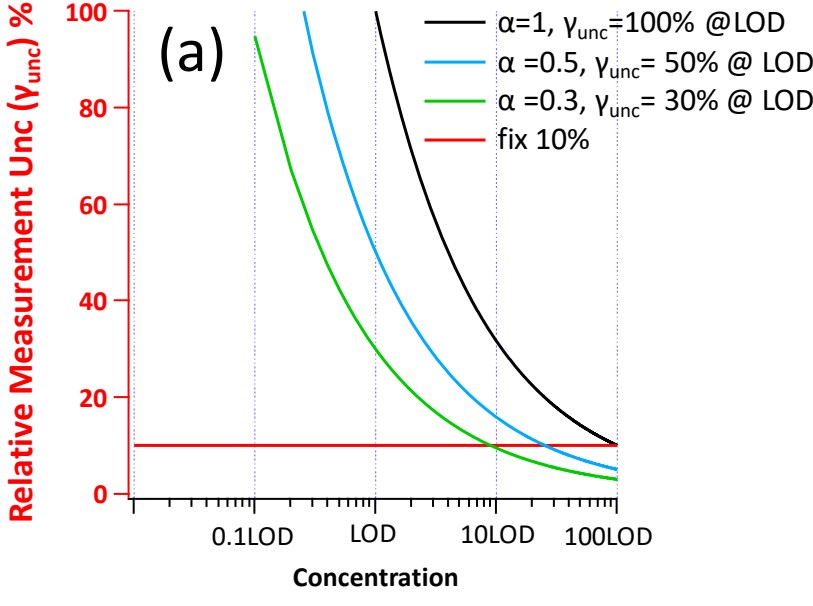

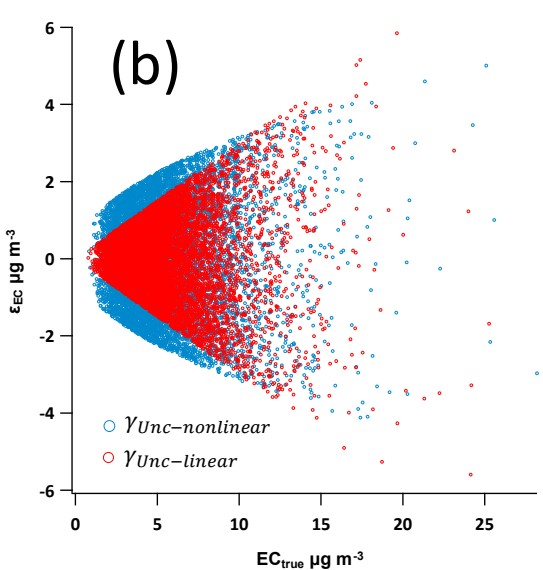


**Figure 1.** (a) Example $\gamma_{Unc-nonlinear}$ curves by different α values (Eq. (17)). The X axis is concentration (normalized by LOD) in log scale and Y axis is $\gamma_{Unc}$. Black, blue and green line represent α equal to 1, 0.5 and 0.3, respectively, corresponding to the $\gamma_{Unc-nonlinear}$ at LOD level equals to 100%, 50% and 30%, respectively. The red line represents $\gamma_{Unc-linear}$ of 10%. (b) Example of measurement uncertainty generation of $\gamma_{Unc-nonlinear}$ and $\gamma_{Unc-linear}$. The blue circles represent $\gamma_{Unc-nonlinear}$ following Eq. (17) ($LOD_{EC} = 1$ , $a_{EC} = 1$). The red circles represent $\gamma_{Unc-linear}$ (30%).



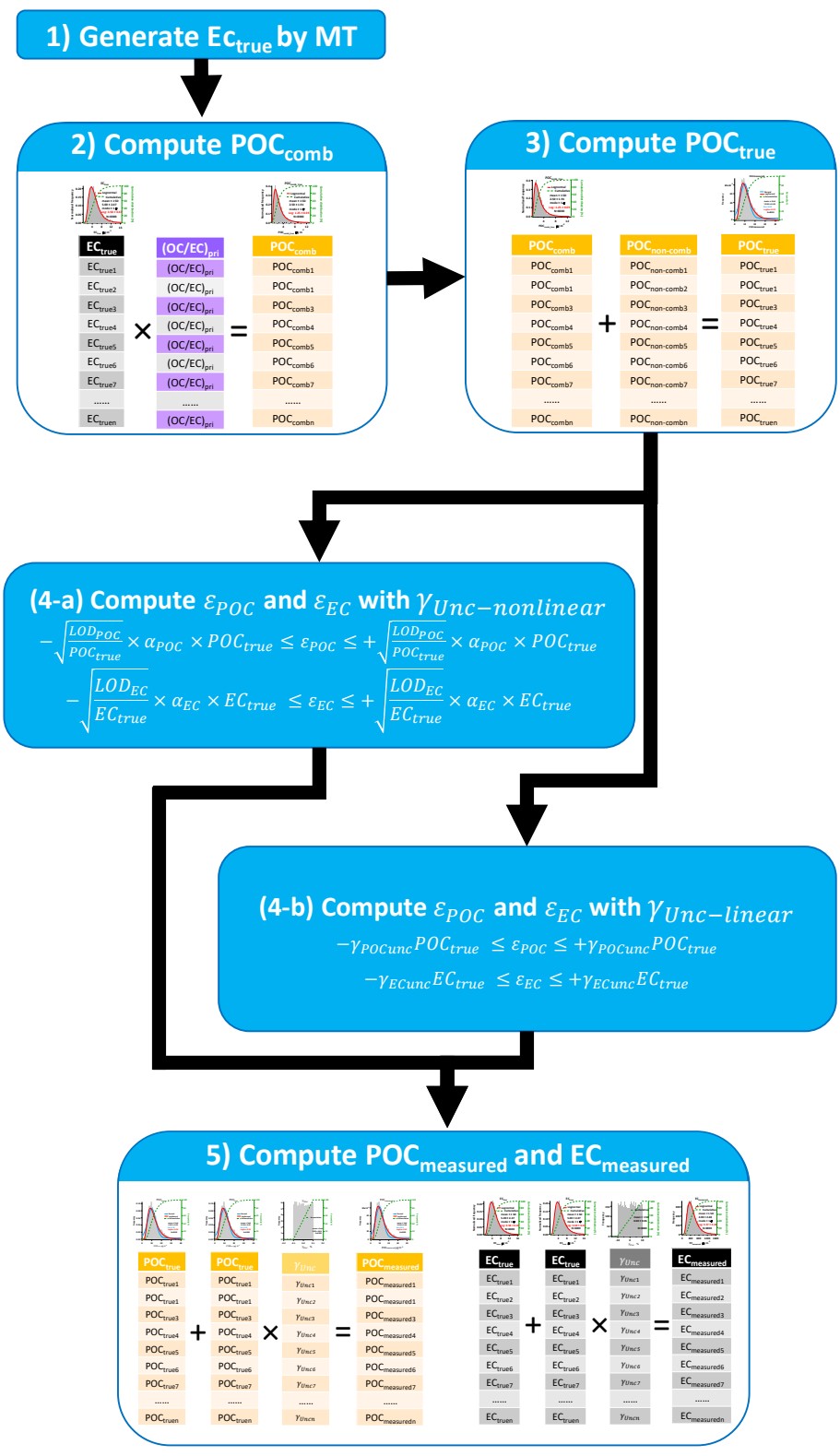


**Figure 2.** Flowchart of data generation steps using MT.


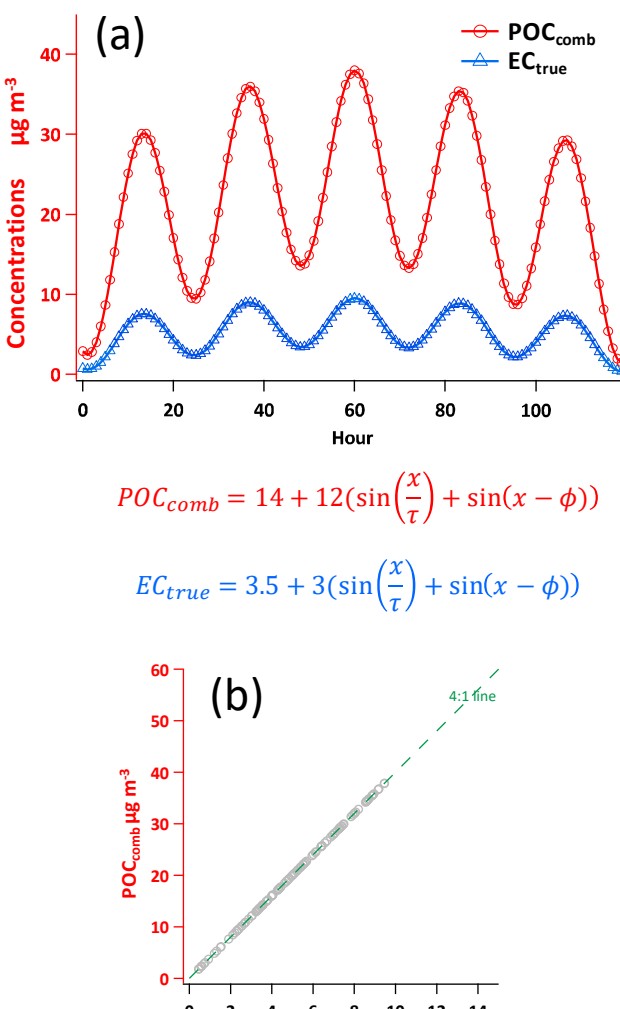

$$POC_{comb} = 14 + 12(\sin\left(\frac{x}{\tau}\right) + \sin(x - \phi))$$

$$EC_{true} = 3.5 + 3(\sin\left(\frac{x}{\tau}\right) + \sin(x - \phi))$$


**Figure 3.** $POC_{comb}$ and $EC_{trure}$ data generated by the sine functions of Chu (2005). (a) Time series of the 120 data points for $POC_{comb}$ and $EC_{true}$. (b) Scatter plot of $POC_{comb}$ vs. $EC_{true}$



# Comparison study design

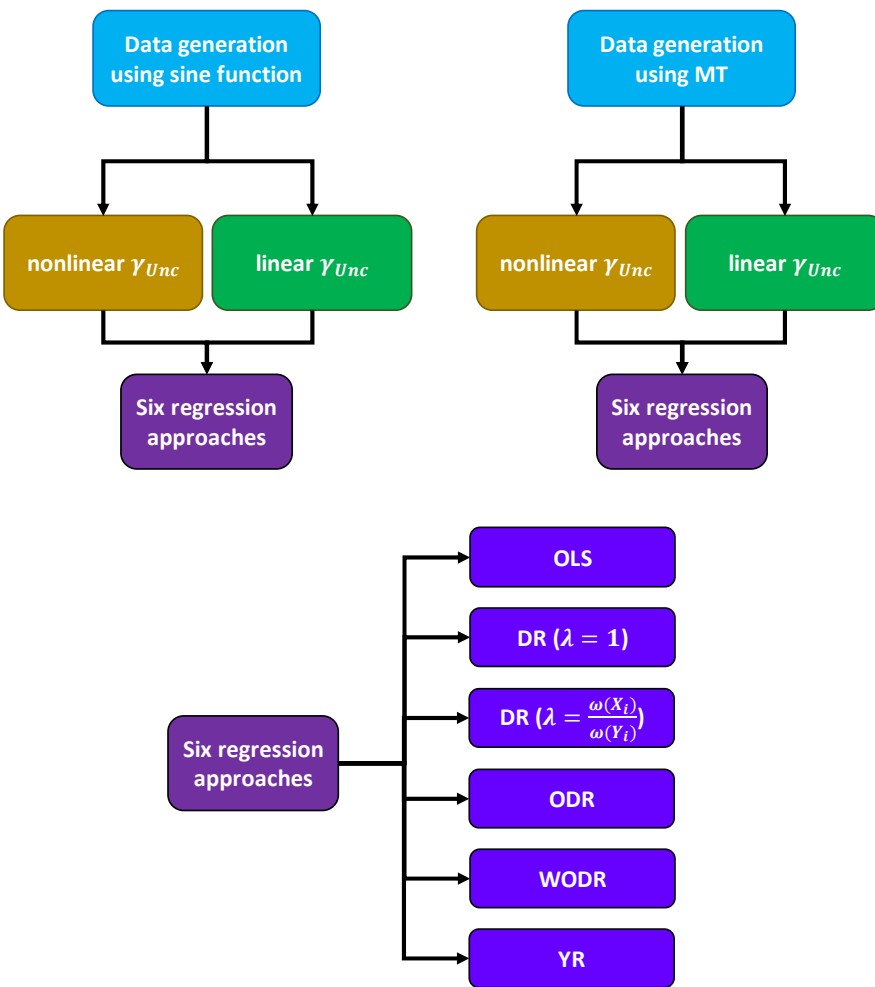

**Figure 4.** Overview of the comparison study design.

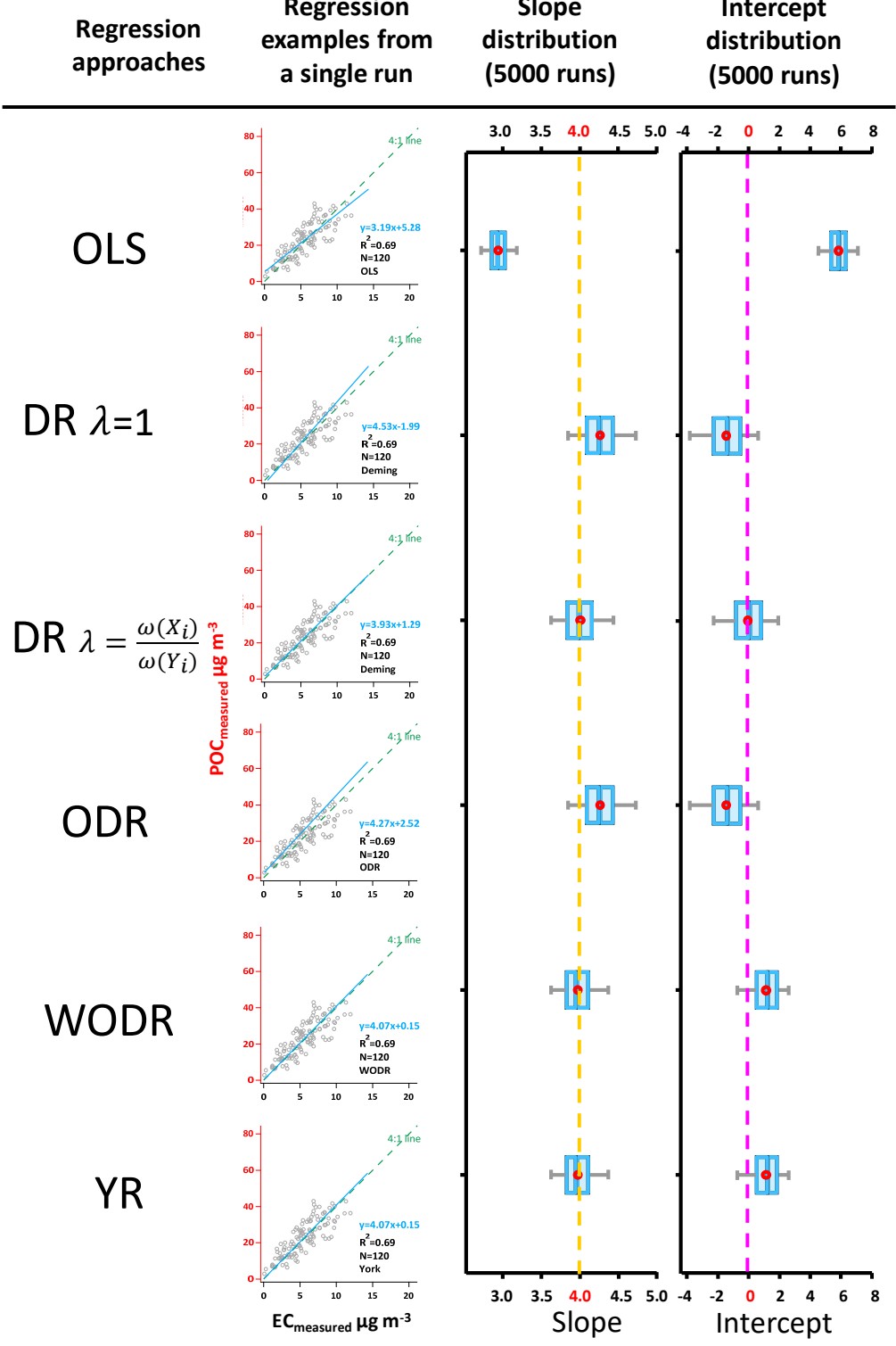


**Figure 5.** Regression results on synthetic data, case 1 (Slope=4, Intercept=0, $LOD_{POC}$=1, $LOD_{EC}$=1, $a_{POC}$=1, $a_{EC}$=1, $R^2$ (POC, EC) =0.67±0.03). The scatter plots demonstrate regression examples from a single run. The box plots show the distribution of regressed slopes and intercepts from 5000 runs of six regression approaches. The dashed line in orange and peachblow represent true slope and intercept, respectively.

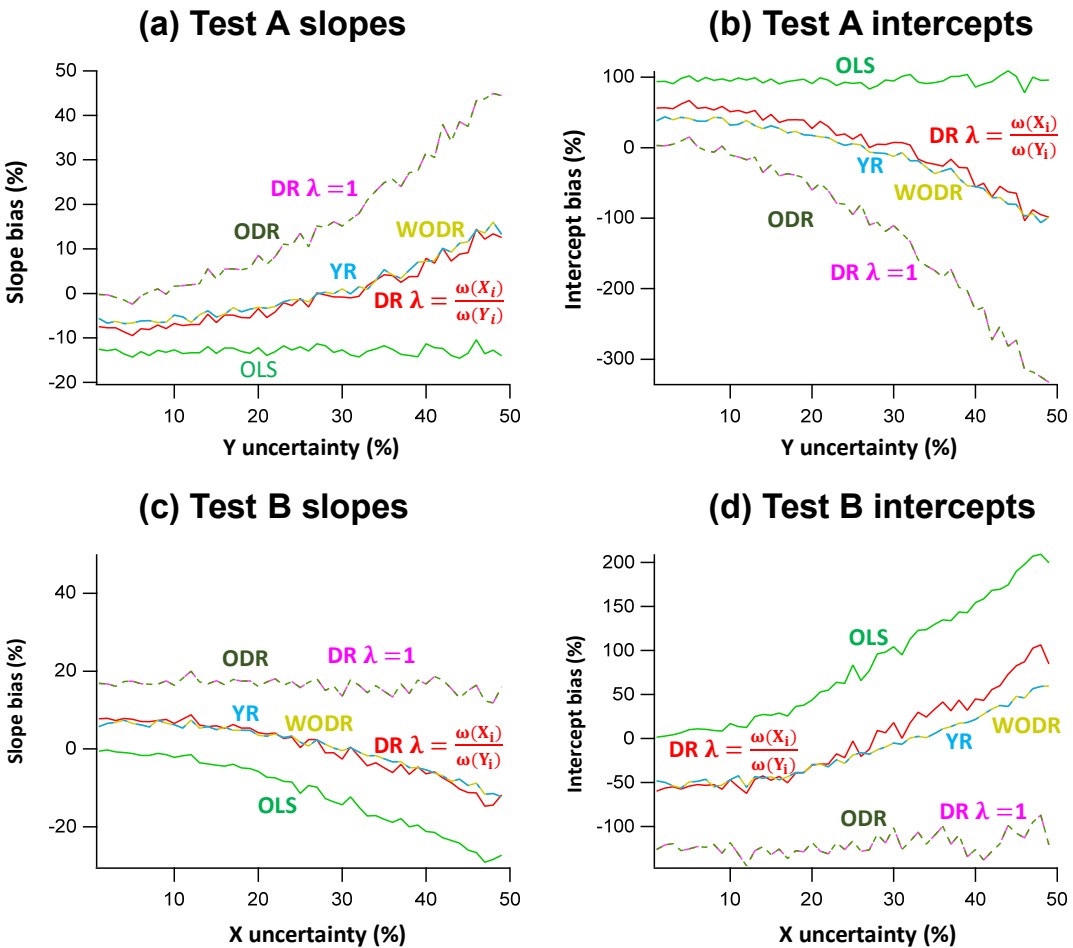


**Figure 6.** Slope and intercept biases by different regression schemes in two test scenarios (A
and B) in which the assumed error for one of the regression variables deviates from the actual
measurement error. In Test A data generation, $\gamma_{Unc\_X}$ is fixed at 30% and $\gamma_{Unc\_Y}$ is varied
between 1 ~ 50%. In Test B, $\gamma_{Unc\_X}$ varies between 1 ~ 50% and $\gamma_{Unc\_Y}$ is fixed at 30%. The
"true" measurement error for regression is 10% for both X and Y. (a) Slopes biases as a function
of $\gamma_{Unc\_Y}$ in Test A. (b) Intercepts biases as a function of $\gamma_{Unc\_Y}$ in Test A. (c) Slopes biases as
a function of $\gamma_{Unc\_X}$ in Test B. (d) Intercepts biases as a function of $\gamma_{Unc\_X}$ in Test B.

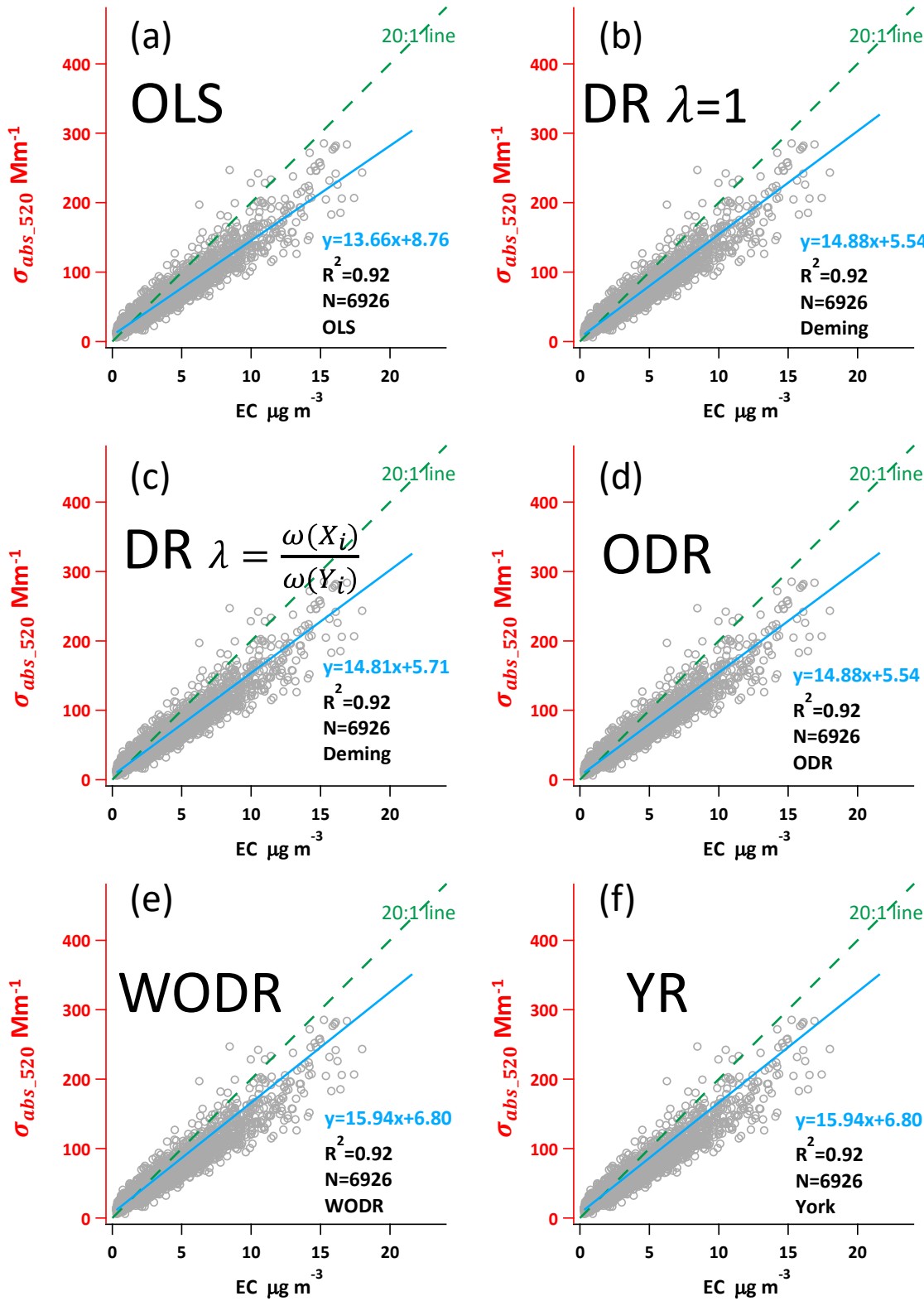


**Figure 7.** Regression results using ambient $\sigma_{abs520}$ and EC data from a suburban site in

Guangzhou, China.

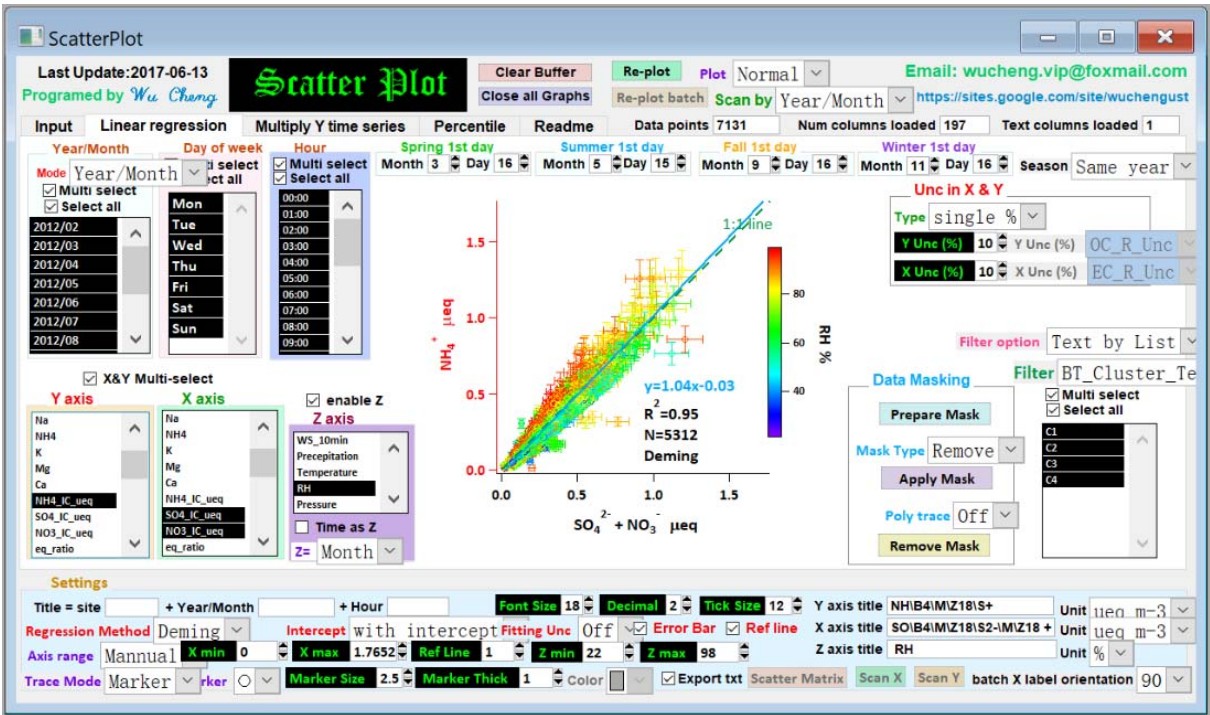

**Figure 8.** The user interface of Scatter Plot Igor program. The program and its operation
manual are available from: https://doi.org/10.5281/zenodo.832417.