# Peer review of "Evaluation of linear regression techniques for"

_Atmospheric Measurement Techniques, 2017_

## Referee Comment (RC1) · Anonymous Referee #1 · 29 Sep 2017

The paper is an extension of the work by Saylor et al. (2006) and shows that ordinary least squares (OLS) techniques are not the best techniques in comparing two variables which both have errors in measurements.

The paper is well written and the science is good.

However, one can discuss the 'new science' of the paper. What is discussed in the paper, that OLS is a flawed method for comparing variables with errors, should be known to many researchers. However, reviewing the literature, one can see that it is not as widely known as it should be. Indeed, the OLS is often still abused in literature. Therefore, if this paper manages to increase the knowledge in using better regression

methods for these cases, it will have served it purpose. As a result, despite the lack of a lot of 'new science', I would still accept the paper, albeit when another case that is lacking now is discussed. Discussion of this case would improve the usefullness of this paper strongly in my opinion: OLS is still widely used when comparing for instance model and measurement data. It would be interesting to add such a case, where the a priori error in one of the variables is unknown. What regression techniques would then be ideal? This can happen too with measurement techniques, if for instance, the technical errors of a measurement described cannot be trusted. And what is the best technique if the errors on both the independent and the dependent variable are unknown? How to proceed in that case?

Adding this discussion would, in my opinion, improve the manuscript.

Technical point: Last sentence of §3.1.2: meaning of SI?

---

## Referee Comment (RC2) · Anonymous Referee #2 · 26 Oct 2017

Review of "Evaluation of linear regression techniques for atmospheric applications: The importance of appropriate weighting" by Wu and Yu

General comments:
This manuscript evaluates five linear regression techniques, ranging from standard (ordinary) least squares to those that account for errors in both variables. Described is a technique to generate data with desired properties for analysis by the regression techniques. The proper accounting for uncertainties, and thus the appropriate weighting, is emphasized. Approaches are recommended that retrieve slopes and intercepts of datasets with uncertainties in x and y variables that have minimal bias in slope and intercept.

The analysis is systematic and apparently carefully done. It does surprise this reviewer, however, that none of the regression techniques precisely recover the input slope and intercept (for example, results from Figure 5), particularly for the more sophisticated methods. Other papers have shown that the York method retrieves correct slopes and intercepts for a wide variety of conditions. It seems that with 5000 (or more) runs, regression with proper weighting should yield average slopes and intercepts very close to the input values. Suggest making use of Pearson's data with York's weights (for which the slope and intercept are known with high accuracy) to verify the coding used to perform the regression, as there may be some coding errors that remain and are affecting the results. The coding for data generation should also be checked carefully to ensure that this is not the problem. It is just stated that the $r^2$ value is 0.67. The situation at the top of Figure S2 is what I would expect for properly generated data with proper accounting for uncertainties in x and y, namely that the average slope and intercept are precisely the input values.

The data generation schemes presented need more explanation. To test data regression schemes, it is not necessary that the data behave precisely like ambient atmospheric data. While not stated, it appears that the Chu 2005) method is attempted to reproduce the diurnal behavior of species concentration. This reviewer does not see that the use of this method adds to the comparison of the various regression methods, and probably only adds confusion. Suggest either providing a better explanation and justification of using this approach, or remove it from the paper.

Specific comments:
Several cases are considered in the paper and the supplemental material. For clarity, suggest presenting all the cases in a single table, showing the input slopes and intercepts as well as the linear and non-linear uncertainties of the x and y variables. Yes, values are shown for some of the cases, but they are split between the main paper and the supplement, and are hard to directly compare. There is also inconsistency between Figure 4, which indicates that there are 12 scenarios, and the various tables that go up to Case 18 (Table S7). Suggest describing the various scenarios in the text earlier in the paper than page 14 where they are discussed.

Page 3, line 57. Suggest "…is much smaller than the uncertainty…". Suggest making this discussion more quantitative. In other words, give a precise value and to the how large the relative uncertainty must be to require use of techniques beyond OLS.

Page 3, line 59. Suggest "…may have comparable degrees of uncertainty."

Page 3, line 61. Suggest "…applied to the dataset."

Page 4, line 78. Suggest "In principle, a best-fit regression line should have greater dependence on the more precise data points rather than the less reliable ones."

Page 4, line 81. Suggest "…is closer to the correct value than OLS, but may…"

Page 4, line 84. Suggest "This $\lambda$ value is the key to handling the…"

Page 4, line 85. Suggest "…for the best-fit line calculations."

Page 4, line 86. Suggest "…in the calculation of the best-fit line for an error-in-variable…"

Page 5, equations 2, 3 and 4. It appears that brackets or parentheses are needed to include both the $x_i$ and $y_i$ containing terms in the summation (such as done in equation 6).

Page 6, line 136. Suggest "…for demonstration in a real-world application."

Page 6, line 146. Not sure why the word "relatively" was added. Suggest removing it.

Page 7, line 166. Suggest "…$POC_{comb}$ (the part of Y that is correlated with X)…"

Page 7, line 168. Suggest "…is added to $POC_{comb}$…"

Page 7, line 178. Suggest "…uncertainties ($\varepsilon_{comb}$) to the true…". Also, suggest indicating (somewhere) that the uncertainties are both positive and negative with a defined distribution, and an average of 0.

Page 8, line 199. The modification of the definition of $\Upsilon_{unc}$ is stated, but no references are given, and the justification is not clear. Does this formula represent the uncertainties in an appropriate way?

Page 9, line 209-211. Related to the previous comment, this is asserted, but not really proven.

Page 9, line 212. Does "uniform distribution" mean "flat distribution" (also used on page 8, line 180)? In other words, is the distribution variance (and thus the weight) constant with deviation from the mean (rather than Gaussian or some other distribution). If so, why was this chosen?

Page 9, equations 20 and 21. The origin of these equations is not clear. Why is $EC_{true}$ multiplied by $LOD_{EC}$? Why is the factor of 3 included?

Page 9, line 223. Suggest "…where $\Upsilon_{POCUnc}$ and $\Upsilon_{ECnc}$ are the relative measurement uncertainties…"

Page 10, line 239. Have you done analyses of the fitting accuracy with various frequency distributions? Since ambient data is typically log-normal distributed, its use might make sense, if it does make a difference.

Page 10, line 253. Suggest "…in this study is a single value…"

Page 11, line 260. It might be useful to have separate symbols for the non-linear and linear parts of the uncertainties (e.g. $\Upsilon_{Unc-linear}$ and $\Upsilon_{Unc-nonlinear}$).

Page 11, line 264. Suggest "…is given in the Supplemental Information."

Page 11, Section 3.1.3. Suggest a statement indicating why Chu (2005) used this method to generate data (if this remains in the paper per earlier comment).

Page 11, line 278. Suggest "…goodness of the regression intercept..".

Page 12, line 291. Suggest "…instruments utilized inlets with a 2.5 µm particle diameter cutoff."

Page 12, line300. Do you mean "SigmaPlot" rather than "Sigma Pro"?  Also suggest "…DR is set to 1…"

Page 15, line 369. Suggest "…unbiased slope…and intercept…"

Page 17, line 426. Suggest "…the results using synthetic data."

Page 17, line 434, 435, and 437. Suggest changing "percentile" to "percentage".

Page 17, line 438. Suggest "…the differences between the six RS are also small…"

Page 17, line 439. Suggest "…as $r^2$ decreases…"

Page 17, line 443. Suggest "…confirm the results obtained in comparing methods with the…"

Page 18, line 455. Suggest "…the measurement errors during…"

Page 18, line 456. Suggest "…data with $r^2$ less than…"

Page 18, line 457. Suggest "…to minimize biases in the slope…"

Page 18, line 461-2. Suggest "…packed with many use features for data analysis and plotting,…"

Page 18, line 464. Is the program planned to be archived at the given site for a long time?  Check the journal's policies regarding links to download sites.

Page 20, line 498. Suggest that for the York iterative method, that a relative tolerance between successive iterations be calculated, and that convergence be considered when this tolerance is reached. While 6 iterations could be sufficient for some datasets, it may not be enough for others.

Page 24.  There are a few abbreviations missing from the table (e.g. $W_i$, $\beta_i$, $r_i$).

Supplemental Material
Page 2, line 20. Suggest "…is often impacted by…'

---

## Author Comment (AC1) · 20 Dec 2017

**Point-by-point response to review comments on manuscript amt-2017-300 "Evaluation of linear regression techniques for atmospheric applications: The importance of appropriate weighting"**

**By Cheng Wu et and Jian Zhen Yu**

We thank the two anonymous reviewers for their constructive comments to improve the manuscript. Our point-by-point responses to the review comments are listed below. Changes to the manuscript are marked in blue in the revised manuscript. The marked manuscript is submitted together with this response document.

**Anonymous Referee #1**

R1-Q1.

The paper is an extension of the work by Saylor et al. (2006) and shows that ordinary least squares (OLS) techniques are not the best techniques in comparing two variables which both have errors in measurements.

The paper is well written and the science is good.

However, one can discuss the 'new science' of the paper. What is discussed in the paper, that OLS is a flawed method for comparing variables with errors, should be known to many researchers. However, reviewing the literature, one can see that it is not as widely known as it should be. Indeed, the OLS is often still abused in literature. Therefore, if this paper manages to increase the knowledge in using better regression methods for these cases, it will have served it purpose. As a result, despite the lack of a lot of 'new science', I would still accept the paper, albeit when another case that is lacking now is discussed. Discussion of this case would improve the usefulness of this paper strongly in my opinion: OLS is still widely used when comparing for instance model and measurement data. It would be interesting to add such a case, where the a priori error in one of the variables is unknown. What regression techniques would then be ideal? This can happen too with measurement techniques, if for instance, the technical errors of a measurement described cannot be trusted. And what is the best technique if the errors on both the independent and the dependent variable are unknown? How to proceed in that case?

Adding this discussion would, in my opinion, improve the manuscript.

**Author's Response:** The reviewer raised a very good point and we fully agree that including corresponding tests would improve the usefulness of the manuscript. To address this question, we added a new section with two tests (Figure R-1) in the manuscript. The corresponding discussion are shown below.

**4.4 Caveats of regressions with unknown X and Y uncertainties**

[revised manuscript text omitted]

Following contents are added to the abstract to cover the findings in section 4.4.

If discrepancy exist between measurement error of data and measurement uncertainty used for regression, DR, WODR and YR can provide the least biases in slope and intercept among all tested regression techniques. For these reasons, DR, WODR and YR are recommended for atmospheric studies when both X and Y data have measurement errors.

The first paragraph of conclusion is updated to reflect the finding in section 4.4.

This study aims to provide a benchmark of commonly used linear regression algorithms using a new data generation scheme (MT). Six regression approaches are tested, including OLS, DR ($\lambda = 1$), DR ($\lambda = \frac{\omega(X_i)}{\omega(Y_i)}$), ODR, WODR and YR.

The results show that OLS fails to estimate the correct slope and intercept when both X and Y have measurement errors. This result is consistent with previous studies. For ambient data with $R^2$ less than 0.9, error-in-variables regression is needed to minimize the biases in slope and intercept. If measurement uncertainties in X and Y are determined during the measurement, measurement uncertainties should be used for regression. With appropriate weighting, DR, WODR and YR can provide the best results among all tested regression techniques. Sensitivity tests also reveal the importance of the weighting parameter $\lambda$ in DR. An improper $\lambda$ could lead to biased slope and intercept. Since the $\lambda$ estimation depends on the form of the measurement errors, it is important to determine the measurement errors during the experimentation stage rather than making assumptions. If measurement errors are not available from the measurement and assumptions are made on measurement errors, DR, WODR and YR are still the best option that can provide the least bias in slope and intercept among all tested regression techniques. For these reasons, DR, WODR and YR are recommended for atmospheric studies when both X and Y data have measurement errors.

**Technical comments**

**R1-Q2.** Last sentence of §3.1.2: meaning of SI?

**Author's Response:** Supplemental information. The sentence had been revised to "
[revised manuscript text omitted]

**1    Comparion of three York regression implementations**

A variety of York regression implementations are compared using the Pearson's data with

York's weights according to York (1966) (abbreviated as "PY data" hereafter). The dataset is Table S1.Three York regression implementations are compared using the PY data, including spreadsheet by Cantrell (2008), Igor program by this study and a commercial software (OriginPro™ 2017). The three York regression implementations yield identical slope and intercept as shown in the highlighted areas (in red) in Figure S5. These crosscheck results suggest that the codes in our Igor program can retrieve consistent slopes and intercepts as other proven programs did.

**2    Impact of two primary sources in OC/EC regression**

A sampling site is often impacted by multiple combustion sources in the real atmosphere.

[revised manuscript text omitted]

MT Igor program can be downloaded from the following link:

https://sites.google.com/site/wuchengust.

---

## Author Comment (AC2) · 20 Dec 2017

**Point-by-point response to review comments on manuscript amt-2017-300 "Evaluation of linear regression techniques for atmospheric applications: The importance of appropriate weighting"**

**By Cheng Wu et and Jian Zhen Yu**

We thank the two anonymous reviewers for their constructive comments to improve the manuscript. Our point-by-point responses to the review comments are listed below. Changes to the manuscript are marked in blue in the revised manuscript. The marked manuscript is submitted together with this response document.

**Anonymous Referee #2**

**General Comments**

**R2-Q1.** This manuscript evaluates five linear regression techniques, ranging from standard (ordinary) least squares to those that account for errors in both variables. Described is a technique to generate data with desired properties for analysis by the regression techniques. The proper accounting for uncertainties, and thus the appropriate weighting, is emphasized. Approaches are recommended that retrieve slopes and intercepts of datasets with uncertainties in x and y variables that have minimal bias in slope and intercept.

The analysis is systematic and apparently carefully done. It does surprise this reviewer, however, that none of the regression techniques precisely recover the input slope and intercept (for example, results from Figure 5), particularly for the more sophisticated methods. Other papers have shown that the York method retrieves correct slopes and intercepts for a wide variety of conditions. It seems that with 5000 (or more) runs, regression with proper weighting should yield average slopes and intercepts very close to the input values. Suggest making use of Pearson's data with York's weights (for which the slope and intercept are known with high accuracy) to verify the coding used to perform the regression, as there may be some coding errors that remain and are affecting the results. The coding for data generation should also be checked carefully to ensure that this is not the problem. It is just stated that the $r^2$ value is 0.67. The situation at the top of Figure S2 is what I would expect for properly generated data with proper accounting for uncertainties in x and y, namely that the average slope and intercept are precisely the input values.

**Author's Response:** Thanks for the suggestion. A variety of York regression implementations are compared using the Pearson's data with York's weights according to York (1966) (abbreviated as "PY data" hereafter). The dataset is shown below in Table R-1.

**Table R-1.** Pearson's data with York's weights according to York (1966).

| $X_i$ | $\omega(X_i)$ | $Y_i$ | $\omega(Y_i)$ |
|-------|---------------|-------|---------------|
| 0 | 1000 | 5.9 | 1 |
| 0.9 | 1000 | 5.4 | 1.8 |
| 1.8 | 500 | 4.4 | 4 |
| 2.6 | 800 | 4.6 | 8 |
| 3.3 | 200 | 3.5 | 20 |
| 4.4 | 80 | 3.7 | 20 |
| 5.2 | 60 | 2.8 | 70 |
| 6.1 | 20 | 2.8 | 70 |
| 6.5 | 1.8 | 2.4 | 100 |
| 7.4 | 1 | 1.5 | 500 |

Three York regression implementations are compared using the PY data, including spreadsheet by Cantrell (2008), Igor program by this study and a commercial software (OriginPro™ 2017). The three York regression implementations yield identical slope and intercept as shown in the highlighted areas (in red) in Figure R-2. These crosscheck results suggest that the codes in our Igor program can retrieve consistent slopes and intercepts as other proven programs did.

The coding for MT data generation had been checked carefully and it works as expected. One evidence is that DR ($\lambda = \frac{\omega(X_i)}{\omega(Y_i)}$) can successfully retrieve unbiased slope and intercept as shown in the new Table 2. YR exhibit small biases for some non-zero true intercept cases. The intercept retrieve accuracy of YR is not as good as DR ($\lambda = \frac{\omega(X_i)}{\omega(Y_i)}$) for some cases shown in Table 2. But the degree of intercept bias is still acceptable. It is also worth noting that all cases shown in Table 2 is based on a situation that measurement uncertainty used for regression truly represent the measurement uncertainty from data generation. This might not always be the case in the real-world application. Working on ambient data could easily encounter inconsistency between the measurement error used for linear regression and measurement error embed in the data. The results for such cases are discussed in the section 4.4 of revised manuscript (also shown in the response to R1-Q1). Results from these cases suggest that YR can provide similar slope and intercept retrieval accuracy as DR ($\lambda = \frac{\omega(X_i)}{\omega(Y_i)}$) did. In this sense, despite the slightly biases observed in some cases shown in Table 2, YR is still recommended for ambient data application.

[Figure]

[Figure]

[Figure]

**Figure R-2.** York regression implementations comparison, including spreadsheet by Cantrell (2008), Igor program by this study and a commercial software (OriginPro™ 2017).

**R2-Q2.** The data generation schemes presented need more explanation. To test data regression schemes, it is not necessary that the data behave precisely like ambient atmospheric data. While not stated, it appears that the Chu 2005) method is attempted to reproduce the diurnal behavior of species concentration. This reviewer does not see that the use of this method adds to the comparison of the various regression methods, and probably only adds confusion. Suggest either providing a better explanation and justification of using this approach, or remove it from the paper.

**Author's Response:** The inclusion of Chu2005 data generation scheme mainly serves two purposes. First, this scheme is adopted by two previous studies (Chu, 2005;Saylor et al., 2006), including Chu2005 data generation scheme can help to verify whether the regression codes in Igor can reproduce the results from the two previous studies. Second, consistency check between results from Chu2005 and MT provides a circumstantial evidence that the MT code works as expected. We add following contents in section 3.1.3 to justify the inclusion of Chu2005 data generation.

> Beside MT, the inclusion of the sine function data generation schemes in this study mainly serves two purposes. First, the sine function scheme had been adopted by two previous studies (Chu, 2005;Saylor et al., 2006), the inclusion of this scheme can help to verify whether the codes in Igor for various regression approaches can yield the same results from the two previous studies. Second, crosscheck between results from sine function and MT can provides circumstantial evidence that the MT scheme works as expected.

**Specific comments:**

**R2-Q3.** Several cases are considered in the paper and the supplemental material. For clarity, suggest presenting all the cases in a single table, showing the input slopes and intercepts as well as the linear and nonlinear uncertainties of the x and y variables. Yes, values are shown for some of the cases, but they are split between the main paper and the supplement, and are hard to directly compare. There is also inconsistency between Figure 4, which indicates that there are 12 scenarios, and the various tables that go up to Case 18 (Table S7). Suggest describing the various scenarios in the text earlier in the paper than page 14 where they are discussed.

**Author's Response:** We agreed that combining tables into one would improve clarity and minimize the information fragmentation. Result from 18 cases have been integrated into the new Table 2 in the revised manuscript. And the new table is also shown here as Table R-2. The term "six regression scenarios" has been changed to "six regression approaches" to avoid confusion from the 18 cases. We add the following descriptions in section 4 to mention there will be 18 cases discussed.

[revised manuscript text omitted]

**R2-Q4.** Page 3, line 57. Suggest "…is much smaller than the uncertainty…". Suggest making this discussion more quantitative. In other words, give a precise value and to the how large the relative uncertainty must be to require use of techniques beyond OLS.

**Author's Response:** Suggestion taken. The corresponding content has been revised as follows:

The uncertainty of gravimetric analysis is typically less than 1% (Lacey and Faulkner, 2015), which is much smaller compared to the uncertainty of the instrument response. Thus, the error free assumption in "X" is fulfilled.

**R2-Q5.** Page 3, line 59. Suggest "…may have comparable degrees of uncertainty."
**R2-Q6.** Page 3, line 61. Suggest "…applied to the dataset."
**R2-Q7.** Page 4, line 78. Suggest "In principle, a best‑fit regression line should have greater dependence on the more precise data points rather than the less reliable ones."
**R2-Q8.** Page 4, line 81. Suggest "…is closer to the correct value than OLS, but may…"
**R2-Q9.** Page 4, line 84. Suggest "This λ value is the key to handling the…"
**R2-Q10.** Page 4, line 85. Suggest "⋯for the best‑fit line calculations."
**R2-Q11.** Page 4, line 86. Suggest "⋯in the calculation of the best‑fit line for an error‑in‑variable⋯"

**Author's Response:** Revisions made.

**R2-Q12.** Page 5, equations 2, 3 and 4. It appears that brackets or parentheses are needed to include both the $x_i$ and $y_i$ containing terms in the summation (such as done in equation 6).

**Author's Response:** Brackets are added in equations 2, 3 and 4.

**R2-Q13.** Page 6, line 136. Suggest "⋯for demonstration in a real‑world application."
**R2-Q14.** Page 6, line 146. Not sure why the word "relatively" was added. Suggest removing it.
**R2-Q15.** Page 7, line 166. Suggest "…$POC_{comb}$ (the part of Y that is correlated with X)…"
**R2-Q16.** Page 7, line 168. Suggest "…is added to $POC_{comb}$..."

**Author's Response:** Revisions made.

**R2-Q17.** Page 7, line 178. Suggest "…uncertainties (εcomb) to the true…". Also, suggest indicating (somewhere) that the uncertainties are both positive and negative with a defined distribution, and an average of 0.

**Author's Response:** Revision made. The sentence following Eq. (12) is revised as:

Here $\varepsilon_{Conc.}$ is the random error following an even distribution with an average of 0,

**R2-Q18.** Page 8, line 199. The modification of the definition of $\Upsilon$unc is stated, but no references are given, and the justification is not clear. Does this formula represent the uncertainties in an appropriate way?

**Author's Response:** We constructed $\gamma_{Unc}$ as stated in equation 17 to represent a situation that $\gamma_{Unc}$ is not constant but concentration depended. Previous studies had demonstrated the dependency of measurement uncertainty on concentration (Thompson and Howarth, 1973;Thompson, 1988;Lee and Ramsey, 2001). The form of concentration dependent $\gamma_{Unc}$ is not necessarily exactly the same as the real-world situation, but we believe equation 17 can reasonably capture the main characteristic of concentration dependent $\gamma_{Unc}$. The corresponding reference has been added in the revised manuscript.

**R2-Q19.** Page 9, line 209-211. Related to the previous comment, this is asserted, but not really proven.

**Author's Response:** "improved" had been changed to "modified".

**R2-Q20.** Page 9, line 212. Does "uniform distribution" mean "flat distribution" (also used on page 8, line 180)? In other words, is the distribution variance (and thus the weight) constant with deviation from the mean (rather than Gaussian or some other distribution). If so, why was this chosen?

**Author's Response:** Yes, uniform distribution refers to "flat distribution". For a uniform distribution in the interval [a,b], the variance is $\frac{1}{12}(a-b)^2$. Uniform distribution had been used in previous studies (Cox et al., 2003;Chu, 2005;Saylor et al., 2006) and is adopted in this study to parameterize measurement error.

**R2-Q21.** Page 9, equations 20 and 21. The origin of these equations is not clear. Why is ECtrue multiplied by LOD$_{EC}$? Why is the factor of 3 included?

**Author's Response:** We add following explanations in the revised manuscript.

For a uniform distribution in the interval [a,b], the variance is $\frac{1}{12}(a-b)^2$. Since $\varepsilon_{POC}$ and $\varepsilon_{EC}$ follows a uniform distribution in the interval as given by Eqs. (18) and (19), the weights in DR and YR (inverse of variance) become:

**R2-Q22.** Page 9, line 223. Suggest "…where ΥPOCUnc and ΥECnc are the relative measurement uncertainties…"

**Author's Response:** Revision made.

**R2-Q23.** Page 10, line 239. Have you done analyses of the fitting accuracy with various frequency distributions? Since ambient data is typically log‑normal distributed, its use might make sense, if it does make a difference.

**Author's Response:** For the performance of the MT pseudorandom number generator, we conduct Kolmogorov–Smirnov (K-S) tests on the generated data for 5000 runs. In Igor Pro's K-S test, two values (D and C) are compared to evaluate if the data passes the test. D represents the K-S statistic, C represents the K-S critical value. If D<C, the samples follow the corresponding distribution (e.g., log-normal distribution). The result shows that 94.4% data having D small than C (Fig. R-3). Hence, we believe the pseudorandom number generator could produce the data following preset characteristics.

[Figure]

**Figure R-3.** Performance of the MT pseudorandom number generator evaluated by K-S tests. The histogram in grey represents D statistic values in K-S test and the red dashed-line represents C. The dash line in green represents cumulative distribution of D. Data with D<C, i.e., data that strictly follow the log-normal distribution, account for 94.4% in 5000 runs.

**R2-Q24.** Page 10, line 253. Suggest "…in this study is a single value…"

**Author's Response:** Revision made.

**R2-Q25.** Page 11, line 260. It might be useful to have separate symbols for the non-linear and linear parts of the uncertainties (e.g. $\Upsilon_{Unc-linear}$ and $\Upsilon_{Unc-nonlinear}$).

**Author's Response:** $\gamma_{Unc-nonlinear}$ and $\gamma_{Unc-linear}$ are adopted throughout the revised manuscript.

**R2-Q26.** Page 11, line 264. Suggest "…is given in the Supplemental Information."

**Author's Response:** Revision made.

**R2-Q27.** Page 11, Section 3.1.3. Suggest a statement indicating why Chu (2005) used this method to generate data (if this remains in the paper per earlier comment).

**Author's Response:** Please refer to the R2-Q2 for the justification of keeping Chu2005 method.

**R2-Q28.** Page 11, line 278. Suggest "…goodness of the regression intercept.".
**R2-Q29.** Page 12, line 291. Suggest "…instruments utilized inlets with a 2.5 μm particle diameter cutoff."
**R2-Q30.** Page 12, line300. Do you mean "SigmaPlot" rather than "Sigma Pro"? Also suggest "…DR is set to 1…"
**R2-Q31.** Page 15, line 369. Suggest "…unbiased slope…and intercept…"
**R2-Q32.** Page 17, line 426. Suggest "…the results using synthetic data."

**Author's Response:** Revisions made.

**R2-Q33.** Page 17, line 434, 435, and 437. Suggest changing "percentile" to "percentage".

**Author's Response:** "OC/EC percentile" is a widely used term in the EC tracer method community and we believe it would be better to stick to it.

**R2-Q34.** Page 17, line 438. Suggest "…the differences between the six RS are also small…"
**R2-Q35.** Page 17, line 439. Suggest "…as r2 decreases…"
**R2-Q36.** Page 17, line 443. Suggest "…confirm the results obtained in comparing methods with the…"
**R2-Q37.** Page 18, line 455. Suggest "…the measurement errors during…"
**R2-Q38.** Page 18, line 456. Suggest "…data with $r^2$ less than…"
**R2-Q39.** Page 18, line 457. Suggest "…to minimize biases in the slope…"
**R2-Q40.** Page 18, line 461-2. Suggest "…packed with many useful features for data analysis and plotting,…"

**Author's Response:** Revisions made.

**R2-Q41.** Page 18, line 464. Is the program planned to be archived at the given site for a long time? Check the journal's policies regarding links to download sites.

**Author's Response:** The program had been archived at https://doi.org/10.5281/zenodo.832417 and this link is also listed in the "Assets" of AMTD page (as shown below).

[Figure]

https://doi.org/10.5194/amt-2017-300
Discussion papers

| Abstract | **Assets** | Discussion | Metrics |

Research article
Sep 2017

**Evaluation of linear regression techniques for atmospheric applications: The importance of appropriate weighting**

Cheng Wu and Jian Zhen Yu

Review status
This discussion paper is a preprint. It is a manuscript under review for the journal Atmospheric Measurement Techniques (AMT).

**Supplement**
https://doi.org/10.5194/amt-2017-300-supplement

**Model code and software**
Scatter Plot
C. Wu
https://doi.org/10.5281/zenodo.832417

Aethalometer data processor
C. Wu
https://doi.org/10.5281/zenodo.832403

Histbox
C. Wu
https://doi.org/10.5281/zenodo.832405

**R2-Q42.** Page 20, line 498. Suggest that for the York iterative method, that a relative tolerance between successive iterations be calculated, and that convergence be considered when this tolerance is reached. While 6 iterations could be sufficient for some datasets, it may not be enough for others.

**Author's Response:** In our Igor codes for York regression, the convergence condition was set as:

$$\frac{k_{i+1} - k_i}{k_i} < e^{-15}$$

To avoid confusion, the corresponding content was revised to "The calculation is straightforward and usually converged in 10 iterations. For example, the iteration count on the data set of (Chu (2005)) is around 6.".

**R2-Q43.** Page 24. There are a few abbreviations missing from the table (e.g. $W_i$, $\beta_i$, $r_i$).

**Author's Response:** Table 1 was updated.

**R2-Q44.** Supplemental Material. Page 2, line 20. Suggest "…is often impacted by…'

**Author's Response:** Typo corrected.

---

## Author Response (AR2)

**Point-by-point response to review comments on manuscript amt-2017-300 "Evaluation of linear regression techniques for atmospheric applications: The importance of appropriate weighting"**

**By Cheng Wu and Jian Zhen Yu**

**Editor comments to the Author:**

The authors have reasonably addressed the comments of the two anonymous referees and they have modified their manuscript accordingly. However, the comments below should be taken into consideration and several alterations are needed in the main text and the Supplement before the manuscript can be published in AMT.

**Author's Response:** We thank the editor for the constructive comments to improve the manuscript. Our point-by-point responses to the review comments are listed below. Changes to the manuscript are marked in blue in the revised manuscript. The marked manuscript is submitted together with this response document.

**Main text:**

Line 23: Replace "tested are" by "five techniques are".

Line 32: Replace "found an" by "found that an".

Line 33: Replace "leads to" by "lead to".

**Author's Response:** Revisions made.

Line 115: It should be indicated what the "n" in the summation stands for. Then, in line 579 the "n" has become "N". The authors should stick to a single symbol; I suggest to use "N".

**Author's Response:** The use of "N" is adopted throughout the manuscript.

Line 127: A literature reference for "Igor" should already be given here.

Line 135: Replace "and Yi" by "and Yi,".

Line 178: Replace "Eq.(7)" by "Eq. (7)".

Line 188: Replace "are explained in section 3.1.2 and 3.1.3 respectively" by "is explained in sections 3.1.2 and 3.1.3, respectively".

Line 200: Replace "uncertainties relative" by "uncertainty relative".

Line 224: Replace "30% respectively" by "30%, respectively".

Line 234: Replace "had been" by "has been".

Line 237: Replace "follows a" by "follow a".

Line 277: Replace "samples are" by "samples is".

Line 286: Replace "3.1, two" by "3.1.1, two".

Line 290: Replace "X respectively to" by "X, respectively, to".

Line 295: Replace "schemes in" by "scheme in".

Line 296: Replace "had been adopted by two" by "was adopted in two".

Line 320: Replace "X respectively to" by "X, respectively, to".

Line 323: Replace "on the top" by "on top".

**Author's Response:** Revisions made.

Line 332: It is unclear to me what "root" is doing here. Should it not be left out?

**Author's Response:** "root" removed.

Line 333: Replace "computor program" by "computer program".

Line 362: Replace "are summarized" by "is summarized".

Line 372: Replace "obtained, however, results from DR with λ=1 shows" by "obtained; however, results from DR with λ=1 show".

Line 383: Replace "by higher the" by "by a higher".

Line 385: Replace "than Case" by "than in Case".

Line 386: Replace "compare to Case" by "compared to Case".

Line 399: Replace "set to be" by "set to".

Line 403: Replace "underestimates the" by "underestimate the".

Line 433: Replace "report unbiased" by "reports unbiased".

Line 445: Replace "approaches report" by "approach reporting".

Line 455: Replace "many commercial" by "much commercial".

**Author's Response:** Revisions made.

Line 474: Replace "embed in" by "embedded in".

**Author's Response:** Content deleted.

Line 478: Replace "sampler is" by "samplers is".

Line 480: Replace "samples and" by "samplers and".

**Author's Response:** Revisions made.

Line 493: Replace "c&d. which" by "c&d which".

**Author's Response:**  The sentence has been rephrased as follows:

In Test B, $\gamma_{Unc\_Y}$ is fixed at 30% and $\gamma_{Unc\_X}$ varies between 1 ~ 50%. The results of Test B are shown in Figs. 6 c and d.

Line 496: Replace "A which" by "A in which".

Line 501: Replace "independent to" by "independent on".

Line 509: Replace "are smaller" by "is smaller".

Line 510: Replace "compare to" by "compared to".

Line 535: Replace "resulting decreased" by "resulting in decreased".

Line 570: Replace "It packed" by "It is packed".

**Author's Response:** Revisions made.

Pages 24-27, References: Titles of journal articles should be in lower case instead of in Title Case. Furthermore, abbreviated journal names should be used; thus in line 657 "Measurement Techniques" should be replaced by "Meas. Tech.".

**Author's Response:** References updated accordingly.

Line 757: Replace on two occasions "that adjust the" by "that adjusts the".

Line 757: Replace "weight orthogonal" by "weighted orthogonal".

Line 766: Replace "0.3 respectively" by "0.3, respectively".

Line 767: Replace "30% respectively" by "30%, respectively".

Line 772: in the top line of Figure 2 replace "generations steps" by "generation steps".

Line 777: Replace "of (Chu (2005))" by "of Chu (2005)".

Line 790: Replace "intercept respectively" by "intercept, respectively".

Line 794: Replace on the second occurrence "varied between" by "is varied between".

**Author's Response:** Revisions made.

**Supplement:**

Line 21: "York (1966)" is not in the list of References.

**Author's Response:** Reference added.

Line 22: Replace "is Table" by "is given in Table".

**Author's Response:** Revision made.

Line 23: "Cantrell (2008)" is not in the list of References.

**Author's Response:** Reference added.

Line 30: Replace "2 we" by "2 of the main text we".

Line 31: Replace "lower than" by "lower than that of".

Line 37: Abbreviations and acronyms, here "ROA", should be defined (written full-out) when first used.

Line 38: Replace "two source" by "two sources".

Line 39: Replace "varis by" by "varies by".

Line 40: Replace "for two" by "for the two".

Line 41: Replace "listted in Table S8" by "listed in Table S2".

**Author's Response:** Revisions made.

Line 46: It is unclear what is meant by "ratio of average"; average of what to average of what? Furthermore, it should be "ratio of averages" instead of "ratio of average".

**Author's Response:** The sentence has been revised to "ratio of averages (ROA here refers to the ratio of averaged OC to averaged EC, which is considered as the true value of slope when intercept=0)".

Line 50: Replace "underestimate the" by "underestimates the".

Line 70: Replace "which consider" by "which considers".

Line 71: Replace "It packed" by "It is packed".

**Author's Response:** Revisions made.

Line 94: It is unclear what is meant by "ratio of averages"; average of what to average of what?

**Author's Response:** The definition of ROA has been revised to "ratio of averages (Y to X, e.g., averaged OC to averaged EC).

Line 96: in the top line of Figure S1 replace "generations steps" by "generation steps".

Line 119: Replace "secnario" by "scenario".

**Author's Response:** Revisions made.

[revised manuscript text omitted]

**1    Comparison of three York regression implementations**

A variety of York regression implementations are compared using the Pearson's data with
York's weights according to York (1966) (abbreviated as "PY data" hereafter). The dataset
is given in Table S2.Three York regression implementations are compared using the PY
data, including spreadsheet by Cantrell (2008), Igor program by this study and a
commercial software (OriginPro™ 2017). The three York regression implementations
yield identical slope and intercept as shown in the highlighted areas (in red) in Figure S6.
These crosscheck results suggest that the codes in our Igor program can retrieve consistent
slopes and intercepts as other proven programs did.

**2    Impact of two primary sources in OC/EC regression**

[revised manuscript text omitted]

| Approach | Sum of squared residuals (SSR) | Calculation |
|---|---|---|
| Ordinary least squares (OLS) | $$S = \sum_{i=1}^{N}(y_i - Y_i)^2$$ | close form |
| Orthogonal distance regression (ODR) | $$S = \sum_{i=1}^{N}[(x_i - X_i)^2 + (y_i - Y_i)^2]$$ | iteration |
| Weighted orthogonal distance regression (WODR) | $$S = \sum_{i=1}^{N}[(x_i - X_i)^2 + (y_i - Y_i)^2/\eta]$$ | iteration |
| Deming regression (DR) | $$S = \sum_{i=1}^{N}[\omega(X_i)(x_i - X_i)^2 + \omega(Y_i)(y_i - Y_i)^2]$$ | close form |
| York regression (YR) | $$S = \sum_{i=1}^{N}\left[\omega(X_i)(x_i - X_i)^2 - 2r_i\sqrt{\omega(X_i)\omega(Y_i)}(x_i - X_i)(y_i - Y_i) + \omega(Y_i)(y_i - Y_i)^2\right]\frac{1}{1 - r_i^2}$$ | iteration |

**Table S2.** Pearson's data with York's weights according to York (1966).

| $X_i$ | $\omega(X_i)$ | $Y_i$ | $\omega(Y_i)$ |
|---|---|---|---|
| 0 | 1000 | 5.9 | 1 |
| 0.9 | 1000 | 5.4 | 1.8 |
| 1.8 | 500 | 4.4 | 4 |
| 2.6 | 800 | 4.6 | 8 |
| 3.3 | 200 | 3.5 | 20 |
| 4.4 | 80 | 3.7 | 20 |
| 5.2 | 60 | 2.8 | 70 |
| 6.1 | 20 | 2.8 | 70 |
| 6.5 | 1.8 | 2.4 | 100 |
| 7.4 | 1 | 1.5 | 500 |

**Table S3.** Abbreviations used in two primary sources study.

| Abbreviation | Definition |
|---|---|
| $EC_1, EC_2$ | EC from source 1 and source 2 in the two sources scenario |
| $f_{EC1}$ | fraction of EC from source 1 to the total EC |
| ROA | ratio of averages (Y to X, e.g., averaged OC to averaged EC) |
| $\gamma\_pri$ | ratio of the $(OC/EC)_{pri}$ of source 2 to source 1 |
| RSD | relative standard deviation |
| $RSD_{EC}$ | RSD of EC |
| $\varepsilon_{EC}$, $\varepsilon_{OC}$ | measurement uncertainty of EC and OC |
| $\gamma_{unc}$ | relative measurement uncertainty |
| $\gamma\_{RSD}$ | the ratio between the RSD values of $(OC/EC)_{pri}$ and EC |

[Figure]

Figure S1. Relationships between data point A and fitting line L. Fitting line by OLS minimize the distance of AB. Fitting line by ODR and DR ($\lambda = 1$) minimize the distance of AC. Fitting line by WODR, DR ($\lambda = \frac{\omega(X_i)}{\omega(Y_i)}$) and YR minimize the distance of AD. AD has a θ degree angle relative to AB and the θ depends on the weights of measurement errors in Y and X.

**Data generation steps by the sine functions of Chu (2005)**

[Figure]

**Figure S2.** Flowchart of data generation steps using the sine functions of Chu (2005).

[Figure]

**Figure S3.** Example of bias in slope and intercept due to improper λ assignment. Data generation: Slope=4, Intercept=0; linear $\gamma_{Unc}$ (30%). (a)&(b) Slopes and intercepts when proper λ is input following linear $\gamma_{Unc}$ $(\lambda = \frac{POC^2}{EC^2})$; (c)&(d) Slopes and intercepts when improper λ is input following non-linear $\gamma_{Unc}$ $(\lambda = \frac{POC}{EC})$.

[Figure]

**Figure S4.** Sensitivity tests of λ calculated by mean, median and mode.

[Figure]

**Figure S5.** Regression slopes as a function of OC/EC percentile. OC/EC percentile range
from 0.5% to 100%, with an interval of 0.5%.

[Figure]

[Figure]

[Figure]

**Figure S6.** York regression implementations comparison, including spreadsheet by Cantrell (2008), Igor program by this study and a commercial software (OriginPro™ 2017).

[Figure]

**Figure S7.** Study of two correlated sources scenario by different $R^2$ between the two sources. (a) $R^2 = 1$ (b) $R^2 = 0.86$ (c) $R^2 = 0.75$ (d) $R^2 = 0.49$

[Figure]

**Figure S8.** Study of two independent sources secnario by different parameters. (a)γ_pri=10, RSD$_{EC1}$=0.2, RSD$_{EC2}$=0.2 (b) γ_pri=10, RSD$_{EC1}$=0.1, RSD$_{EC2}$=0.2 (c) γ_pri=10, RSD$_{EC1}$=0.1, RSD$_{EC2}$=0.1 (d) γ_pri=8, RSD$_{EC1}$=0.1, RSD$_{EC2}$=0.1(e) γ_pri=6, RSD$_{EC1}$=0.1, RSD$_{EC2}$=0.1 (f) γ_pri=4, RSD$_{EC1}$=0.1, RSD$_{EC2}$=0.1

[Figure]

**Figure S9.** MT Igor program. OC and EC data following log-normal distribution can be
generated for statistical study purpose (no time series information). User can define mean
and RSD of EC, (OC/EC)$_{pri}$, SOC/OC ratio, measurement uncertainty, sample size, etc.
MT Igor program can be downloaded from the following link:
https://sites.google.com/site/wuchengust.

---

## Author Response (AR3)

**Point-by-point response to review comments on manuscript amt-2017-300 "Evaluation of linear regression techniques for atmospheric applications: The importance of appropriate weighting"**

**By Cheng Wu and Jian Zhen Yu**

2018-01-21

**Editor comments to the Author:**

The authors have substantially improved the main text and Supplement of their manuscript. However, some alterations are still needed in both before the manuscript can be published in AMT.

**Author's Response:** We thank the editor for the comments to further improve the manuscript. Our point-by-point responses to the review comments are listed below. Changes to the manuscript are marked in blue in the revised manuscript. The marked manuscript is submitted together with this response document.

**Main text:**

Line 116: Replace "e.g., distance" by "i.e., distance".

Line 117: It should be indicated what the "N" in the summation stands for.

Line 122: Replace "e.g., distance" by "i.e., distance".

Lines 128, 130, and 135: Replace "Fig.S1" by "Fig. S1".

Line 150: Replace "Summary of five" by "Summary of the five".

Line 157: Replace "Another two" by "Two other".

Line 468: Replace "many commercial" by "much commercial".

Line 548: Replace "resulting decreased" by "resulting in decreased".

Line 806: Replace "0.3 respectively" by "0.3, respectively".

**Author's Response:** Revisions made.

**Supplement:**

Line 96: Replace "Summary of five" by "Summary of the five"

Line 96, in the last column of the Table S1: Replace "close form" by "closed form" on two occasions.

Lines 103-104: Replace "minimize the" by "minimizes the" on three occasions.

Line 128: Replace "secnario" by "scenario".

**Author's Response:** Revisions applied.

[revised manuscript text omitted]

**1    Comparison of three York regression implementations**

A variety of York regression implementations are compared using the Pearson's data with
York's weights according to York (1966) (abbreviated as "PY data" hereafter). The dataset
is given in Table S2.Three York regression implementations are compared using the PY
data, including spreadsheet by Cantrell (2008), Igor program by this study and a
commercial software (OriginPro™ 2017). The three York regression implementations
yield identical slope and intercept as shown in the highlighted areas (in red) in Figure S6.
These crosscheck results suggest that the codes in our Igor program can retrieve consistent
slopes and intercepts as other proven programs did.

**2    Impact of two primary sources in OC/EC regression**

[revised manuscript text omitted]

| Approach | Sum of squared residuals (SSR) | Calculation |
|---|---|---|
| Ordinary least squares (OLS) | $$S = \sum_{i=1}^{N}(y_i - Y_i)^2$$ | closed form |
| Orthogonal distance regression (ODR) | $$S = \sum_{i=1}^{N}[(x_i - X_i)^2 + (y_i - Y_i)^2]$$ | iteration |
| Weighted orthogonal distance regression (WODR) | $$S = \sum_{i=1}^{N}[(x_i - X_i)^2 + (y_i - Y_i)^2/\eta]$$ | iteration |
| Deming regression (DR) | $$S = \sum_{i=1}^{N}[\omega(X_i)(x_i - X_i)^2 + \omega(Y_i)(y_i - Y_i)^2]$$ | closed form |
| York regression (YR) | $$S = \sum_{i=1}^{N}\left[\omega(X_i)(x_i - X_i)^2 - 2r_i\sqrt{\omega(X_i)\omega(Y_i)}(x_i - X_i)(y_i - Y_i) + \omega(Y_i)(y_i - Y_i)^2\right]\frac{1}{1-r_i^2}$$ | iteration |

**Table S2.** Pearson's data with York's weights according to York (1966).

| $X_i$ | $\omega(X_i)$ | $Y_i$ | $\omega(Y_i)$ |
|---|---|---|---|
| 0 | 1000 | 5.9 | 1 |
| 0.9 | 1000 | 5.4 | 1.8 |
| 1.8 | 500 | 4.4 | 4 |
| 2.6 | 800 | 4.6 | 8 |
| 3.3 | 200 | 3.5 | 20 |
| 4.4 | 80 | 3.7 | 20 |
| 5.2 | 60 | 2.8 | 70 |
| 6.1 | 20 | 2.8 | 70 |
| 6.5 | 1.8 | 2.4 | 100 |
| 7.4 | 1 | 1.5 | 500 |

**Table S3.** Abbreviations used in two primary sources study.

| Abbreviation | Definition |
|---|---|
| $EC_1, EC_2$ | EC from source 1 and source 2 in the two sources scenario |
| $f_{EC1}$ | fraction of EC from source 1 to the total EC |
| ROA | ratio of averages (Y to X, e.g., averaged OC to averaged EC) |
| $\gamma\_pri$ | ratio of the $(OC/EC)_{pri}$ of source 2 to source 1 |
| RSD | relative standard deviation |
| $RSD_{EC}$ | RSD of EC |
| $\varepsilon_{EC}, \varepsilon_{OC}$ | measurement uncertainty of EC and OC |
| $\gamma_{unc}$ | relative measurement uncertainty |
| $\gamma\_{RSD}$ | the ratio between the RSD values of $(OC/EC)_{pri}$ and EC |

[Figure]

**Figure S1.** Relationships between data point A and fitting line L. Fitting line by OLS minimizes the distance of AB. Fitting line by ODR and DR ($\lambda = 1$) minimizes the distance of AC. Fitting line by WODR, DR ($\lambda = \frac{\omega(X_i)}{\omega(Y_i)}$) and YR minimizes the distance of AD. AD has a θ degree angle relative to AB and the θ depends on the weights of measurement errors in Y and X.

**Data generation steps by the sine functions of Chu (2005)**

[Figure]

**Figure S2.** Flowchart of data generation steps using the sine functions of Chu (2005).

[Figure]

**Figure S3.** Example of bias in slope and intercept due to improper λ assignment. Data generation: Slope=4, Intercept=0; linear $\gamma_{Unc}$ (30%). (a)&(b) Slopes and intercepts when proper λ is input following linear $\gamma_{Unc}$  $(\lambda = \frac{POC^2}{EC^2})$; (c)&(d) Slopes and intercepts when improper λ is input following non-linear $\gamma_{Unc}$  $(\lambda = \frac{POC}{EC})$.

[Figure]

**Figure S4.** Sensitivity tests of λ calculated by mean, median and mode.

[Figure]

**Figure S5.** Regression slopes as a function of OC/EC percentile. OC/EC percentile range
from 0.5% to 100%, with an interval of 0.5%.

[Figure]

[Figure]

[Figure]

**Figure S6.** York regression implementations comparison, including spreadsheet by Cantrell (2008), Igor program by this study and a commercial software (OriginPro® 2017).

[Figure]

**Figure S7.** Study of two correlated sources scenario by different $R^2$ between the two sources. (a) $R^2 = 1$ (b) $R^2 = 0.86$ (c) $R^2 = 0.75$ (d) $R^2 = 0.49$.

[Figure]

**Figure S8.** Study of two independent sources secnario by different parameters. (a) $\gamma\_pri=10$, $RSD_{EC1}=0.2$, $RSD_{EC2}=0.2$ (b) $\gamma\_pri=10$, $RSD_{EC1}=0.1$, $RSD_{EC2}=0.2$ (c) $\gamma\_pri=10$, $RSD_{EC1}=0.1$, $RSD_{EC2}=0.1$ (d) $\gamma\_pri=8$, $RSD_{EC1}=0.1$, $RSD_{EC2}=0.1$ (e) $\gamma\_pri=6$, $RSD_{EC1}=0.1$, $RSD_{EC2}=0.1$ (f) $\gamma\_pri=4$, $RSD_{EC1}=0.1$, $RSD_{EC2}=0.1$.

[Figure]

Figure S9. MT Igor program. OC and EC data following log-normal distribution can be generated for statistical study purpose (no time series information). User can define mean and RSD of EC, $(OC/EC)_{pri}$, SOC/OC ratio, measurement uncertainty, sample size, etc. MT Igor program can be downloaded from the following link: https://sites.google.com/site/wuchengust.

---

## Author Response (AR4)

**Point-by-point response to review comments on manuscript amt-2017-300 "Evaluation of linear regression techniques for atmospheric applications: The importance of appropriate weighting"**

**By Cheng Wu and Jian Zhen Yu**

2018-02-06

**Editor comments to the Author:**

The following alterations are still needed before the manuscript can be published in AMT.

**Author's Response:** We thank the editor for the comments to further improve the manuscript. Our point-by-point responses to the review comments are listed below. Changes to the manuscript are marked in blue in the revised manuscript. The marked manuscript is submitted together with this response document.

**Main text:**

Line 119: Replace "that used" by "that is used".

**Author's Response:** Revision made.

**Supplement:**

Line 129: Replace "secnario" by "scenario".

**Author's Response:** Typo corrected.

[revised manuscript text omitted]

**1    Comparison of three York regression implementations**

A variety of York regression implementations are compared using the Pearson's data with

York's weights according to York (1966) (abbreviated as "PY data" hereafter). The dataset is given in Table S2.Three York regression implementations are compared using the PY

data, including spreadsheet by Cantrell (2008), Igor program by this study and a commercial software (OriginPro™ 2017). The three York regression implementations yield identical slope and intercept as shown in the highlighted areas (in red) in Figure S6.

These crosscheck results suggest that the codes in our Igor program can retrieve consistent slopes and intercepts as other proven programs did.

**2    Impact of two primary sources in OC/EC regression**

A sampling site is often influenced by multiple combustion sources in the real atmosphere.

In section 1 and 2 of the main text we evaluate the performance of OLS, DR, WODR and

YR in scenarios of two primary sources and arbitrarily dictate that the $(OC/EC)_{pri}$ of source

1 is lower than that of source 2. By varying $f_{EC1}$ (proportion of source 1 EC to total EC)

from test to test, the effect of different mixing ratios of the two sources can be examined.

Two scenarios are considered (Wu and Yu, 2016): two correlated primary sources and two independent primary sources. Common configurations include: $EC_{total}$=2 µgC m$^{-3}$; $f_{EC1}$

varies from 0 to 100%; ratio of the two $OC/EC_{pri}$ values ($\gamma_{\_pri}$) vary in the range of 2~8.

Studies by Chu (2005) and Saylor et al. (2006) both suggest ratio of averages (ROA) being the best estimator of the expected primary OC/EC ratio when SOC is zeroed. Since the overall $OC/EC_{pri}$ from the two sources varies by $\gamma_{\_pri}$, ROA is considered as the reference

$OC/EC_{pri}$ to be compared with slope regressed by of OLS, DR, WODR and YR. The abbreviations used for the two primary sources study are listed in Table S3.

**2.1    Impact of two correlated primary sources**

Simulations considering two correlated primary sources are performed, to examine the effect on bias in the regression methods. The basic configuration is: $(OC/EC)_{pri1}$=0.5,

$(OC/EC)_{pri2}$=5, $\gamma_{Unc}$=30%, N=8000, intercept=0, and the following terms are compared:

ratio of averages (ROA here refers to the ratio of averaged OC to averaged EC, which is considered as the true value of slope when intercept=0), DR, WODR, WODR' (through origin) and OLS. As shown in Figure S7, when $R^2$ (EC1 vs. EC2) is very high, DR, WODR

and WODR' can provide a result consistent with ROA. If the $R^2$ decreases, the bias of the slope and intercept in DR and WODR is larger. OLS constantly underestimates the slope.

**2.2   Impact of two independent primary sources**

Simulations of two independent primary sources are also conducted. If $RSD_{EC1}=RSD_{EC2}$, slopes and intercepts may be either overestimated or underestimated (Figure S8), and the degree of bias depends on the magnitude of $RSD_{EC1}$ and $RSD_{EC2}$. Larger RSD results in larger bias. Uneven RSD between two sources leads to even more bias (Figure S8 a and b).

The degree of bias also shows dependence on $\gamma\_pri$. If $\gamma\_pri$ decreases, the bias becomes smaller (FigureS8 c~f). These results indicate that the scenario with two independent primary sources poses a challenge to $(OC/EC)_{pri}$ estimation by linear regression.

For the EC tracer method, if EC comes from two primary sources and contribution of the two sources is comparable, the regression slope is no longer suitable for $(OC/EC)_{pri}$

estimation and the subsequent SOC calculation, and making EC a mixture that violates the property of a tracer. For such a situation, pre-separation of EC into individual sources by other tracers (if available) by the Minimum R Squared (MRS) method can provide unbiased

SOC estimation results (Wu and Yu, 2016).

**3   Igor programs for error in variables linear regression and simulated OC**

**EC data generation using MT**

An Igor Pro (WaveMetrics, Inc. Lake Oswego, OR, USA) based program (Scatter plot)

with graphical user interface (GUI) is developed to make the linear regression feasible and user friendly (Figure 8). The program includes Deming and York algorithm for linear regression, which considers uncertainties in both X and Y, that is more realistic for atmospheric applications. It is packed with many useful features for data analysis and plotting, including batch plotting, data masking via GUI, color coding in Z axis, data filtering and grouping by numerical values and strings.

Another program using MT can generate simulated OC and EC concentration through user defined parameters via GUI as shown in Figure S9.

Both Igor programs and their operation manuals can be downloaded from the following links:

https://sites.google.com/site/wuchengust https://doi.org/10.5281/zenodo.832417

**Table S1.** Summary of the five linear regression techniques.

| Approach | Sum of squared residuals (SSR) | Calculation |
|---|---|---|
| Ordinary least squares (OLS) | $$S = \sum_{i=1}^{N}(y_i - Y_i)^2$$ | closed form |
| Orthogonal distance regression (ODR) | $$S = \sum_{i=1}^{N}[(x_i - X_i)^2 + (y_i - Y_i)^2]$$ | iteration |
| Weighted orthogonal distance regression (WODR) | $$S = \sum_{i=1}^{N}[(x_i - X_i)^2 + (y_i - Y_i)^2/\eta]$$ | iteration |
| Deming regression (DR) | $$S = \sum_{i=1}^{N}[\omega(X_i)(x_i - X_i)^2 + \omega(Y_i)(y_i - Y_i)^2]$$ | closed form |
| York regression (YR) | $$S = \sum_{i=1}^{N}\left[\omega(X_i)(x_i - X_i)^2 - 2r_i\sqrt{\omega(X_i)\omega(Y_i)}(x_i - X_i)(y_i - Y_i) + \omega(Y_i)(y_i - Y_i)^2\right]\frac{1}{1 - r_i^2}$$ | iteration |

**Table S2.** Pearson's data with York's weights according to York (1966).

| $X_i$ | $\omega(X_i)$ | $Y_i$ | $\omega(Y_i)$ |
|---|---|---|---|
| 0 | 1000 | 5.9 | 1 |
| 0.9 | 1000 | 5.4 | 1.8 |
| 1.8 | 500 | 4.4 | 4 |
| 2.6 | 800 | 4.6 | 8 |
| 3.3 | 200 | 3.5 | 20 |
| 4.4 | 80 | 3.7 | 20 |
| 5.2 | 60 | 2.8 | 70 |
| 6.1 | 20 | 2.8 | 70 |
| 6.5 | 1.8 | 2.4 | 100 |
| 7.4 | 1 | 1.5 | 500 |

**Table S3.** Abbreviations used in two primary sources study.

| Abbreviation | Definition |
| --- | --- |
| $EC_1, EC_2$ | EC from source 1 and source 2 in the two sources scenario |
| $f_{EC1}$ | fraction of EC from source 1 to the total EC |
| ROA | ratio of averages (Y to X, e.g., averaged OC to averaged EC) |
| $\gamma\_pri$ | ratio of the $(OC/EC)_{pri}$ of source 2 to source 1 |
| RSD | relative standard deviation |
| $RSD_{EC}$ | RSD of EC |
| $\varepsilon_{EC}$, $\varepsilon_{OC}$ | measurement uncertainty of EC and OC |
| $\gamma_{unc}$ | relative measurement uncertainty |
| $\gamma\_{RSD}$ | the ratio between the RSD values of $(OC/EC)_{pri}$ and EC |

[Figure]

**Figure S1.** Relationships between data point A and fitting line L. Fitting line by OLS
minimizes the distance of AB. Fitting line by ODR and DR ($\lambda = 1$) minimizes the distance
of AC. Fitting line by WODR, DR ($\lambda = \frac{\omega(X_i)}{\omega(Y_i)}$) and YR minimizes the distance of AD. AD
has a θ degree angle relative to AB and the θ depends on the weights of measurement errors
in Y and X.

**Data generation steps by the sine functions of Chu (2005)**

[Figure]

**Figure S2.** Flowchart of data generation steps using the sine functions of Chu (2005).

[Figure]

**Figure S3.** Example of bias in slope and intercept due to improper $\lambda$ assignment. Data generation: Slope=4, Intercept=0; linear $\gamma_{Unc}$ (30%). (a)&(b) Slopes and intercepts when proper $\lambda$ is input following linear $\gamma_{Unc}$ ($\lambda = \frac{POC^2}{EC^2}$); (c)&(d) Slopes and intercepts when improper $\lambda$ is input following non-linear $\gamma_{Unc}$ ($\lambda = \frac{POC}{EC}$).

[Figure]

**Figure S4.** Sensitivity tests of λ calculated by mean, median and mode.

[Figure]

**Figure S5.** Regression slopes as a function of OC/EC percentile. OC/EC percentile range
from 0.5% to 100%, with an interval of 0.5%.

[Figure]

[Figure]

[Figure]

**Figure S6.** York regression implementations comparison using data shown in Table S2, including (a) spreadsheet by Cantrell (2008), (b) Igor program by this study and (c) a commercial software (OriginPro® 2017).

[Figure]

**Figure S7.** Study of two correlated sources scenario by different $R^2$ between the two sources. (a) $R^2 = 1$ (b) $R^2 = 0.86$ (c) $R^2 = 0.75$ (d) $R^2 = 0.49$.

[Figure]

**Figure S8.** Study of two independent sources scenario by different parameters. (a) γ_pri=10, RSD$_{EC1}$=0.2, RSD$_{EC2}$=0.2 (b) γ_pri=10, RSD$_{EC1}$=0.1, RSD$_{EC2}$=0.2 (c) γ_pri=10, RSD$_{EC1}$=0.1, RSD$_{EC2}$=0.1 (d) γ_pri=8, RSD$_{EC1}$=0.1, RSD$_{EC2}$=0.1(e) γ_pri=6, RSD$_{EC1}$=0.1, RSD$_{EC2}$=0.1 (f) γ_pri=4, RSD$_{EC1}$=0.1, RSD$_{EC2}$=0.1.

[Figure]

**Figure S9.** MT Igor program. OC and EC data following log-normal distribution can be generated for statistical study purpose (no time series information). User can define mean and RSD of EC, (OC/EC)$_{pri}$, SOC/OC ratio, measurement uncertainty, sample size, etc. MT Igor program can be downloaded from the following link: https://sites.google.com/site/wuchengust.